# Controllability changes pain perception by increasing the precision of expectations

Marie Habermann ✉ & Christian Büchel

The ability to exert control over an intensely unpleasant experience, such as pain, can modulate its perception. It is often assumed that control exerts this modulatory effect through a specific control mechanism. We revisit this issue using a task that allowed participants to either control or predict the intensity of a painful stimulus. By approximating Bayesian perceptual integration with computational models, our data show that acute pain modulation by control can be parsimoniously explained by an increase in expectation precision. Importantly, this effect is present in contrast to a condition in which pain is equally predictable, but not controllable. The control-induced increase in expectation precision leads to activation changes in the periaqueductal gray, the supplementary motor area and the rostral anterior cingulate cortex, regions that mediate the interplay between threat uncertainty, motor-control and descending pain modulation.

Control, the ability to self-induce changes in the environment or one's own behavior, is critical for humans to achieve desirable outcomes and escape undesirable states[1,2]. Control over aversive stimuli is thought to recruit specific safety-signaling pathways to reduce general aversiveness and enable successful learning in future situations[3,4]. Pain is endogenous information that serves as a motivational control signal to protect the body and ensure its functionality[5]. Without the ability to act on and terminate pain or aversive stressors[6,7], only passive coping strategies can be initiated[8] and possibly feelings of helplessness and loss of control emerge[9]. Also, active coping strategies are only possible, if controllability of pain is accurately detected[8]. In addition, pain—as an endogenous behavioral control signal—loses its effectiveness, without the ability to respond to it. This raises an important question: how does control affect the experience of pain?

Various hypotheses have been formulated about the impact of the sense of control on pain: one intuitive theory is that experiencing a loss of control over pain increases the overall unpleasantness of a situation and leads to higher pain ratings[10]. This is in contrast to the informational value theory of pain, which states that the ability to exert control should actually increase pain because the organism can respond by distancing itself from the source of the pain[5].

Previous research has begun to examine the effects of perceived control on the processing of acute pain, with the goal of identifying the specific levels at which this influence manifests. Several studies have shown that higher levels of perceived control can be associated with a reduction in pain intensity[11–16], but others have observed no behavioral effects of control on pain[17–19] or even an increase in pain[20,21].

One possible explanation for the observed heterogeneity of results may be the confounding influence of predictability in many studies. In general, stimuli that are perceived as controllable are also predictable. In such cases, effects attributed to control may also be driven by precise expectations (i.e., predictability)[22]. Recent conceptual advances[23–26] consider pain as the probabilistic integration of expectations with incoming sensory signals. This concept has also been used to explain why, when painful stimuli are unpredictable, their perception is biased toward the average of all different possible intensities[27]. Pain ratings are characterized by the integration of expectations (prior) and sensory inputs (likelihood), each weighted by their respective precision. In the context of unpredictable stimuli, the best-guess expectation is simply the mean, or expected value, of all possible intensity levels, which effectively reduces high intensity stimuli and enhances low intensity stimuli by pulling both toward the mean prior. An unpredictability bias toward the center of an outcome distribution has also been reported in magnitude estimation of auditory or visual stimuli, illustrating common properties in perceptual processes that can be described within a Bayesian framework[28].

In the absence of deception, both predictable and controllable pain provide precise information about the intensity of a stimulus,

Department of Systems Neuroscience, University Medical Center Hamburg-Eppendorf, Hamburg, Germany. ✉ e-mail: habermann.science@gmail.com

which should lead to identical expectations (priors). However, it is conceivable that pain control is associated with an internal increase in the precision of expectations. This more precise expectation could lead to more consistent effects of prior information on perception and could potentially reduce pain more effectively when individuals have control over pain reduction[26], but also lead to perceived enhancement of a controlled high intensity stimulus[21,29]. Explaining the effects of control through expectation would explain its influence on acute pain without the need for an additional control-specific cognitive mechanism.

Many studies have shown that positive and negative expectations of pain recruit the descending pain modulatory system, including the rostral anterior cingulate cortex (rACC)−periaqueductal gray (PAG) pathway to the spinal cord[30–32]. Specifically, the PAG appears to modulate pain in terms of predictability of its intensity[33,34] and integrates sensory input with expectation[35]. The anterior insula has also been repeatedly shown to play a role in the processing of pain-related anticipation[36]. Control over pain has also been associated with activity changes in the anterior insula and the ACC[10,19] and additionally with modulated activity in sensorimotor cortices and posterior insula[12,14,16].

To disentangle the effects of controllability and predictability at the behavioral and neural levels, we developed an experimental paradigm that included three different conditions. In the controllable condition, participants could choose the intensity of a subsequent stimulus. In contrast, in the predictable condition, participants had knowledge of the intensity of the stimuli but no control over it. These conditions were compared to a condition in which the intensity of the upcoming pain was unpredictable and uncontrollable. We then applied different versions of a computational model that approximates Bayesian perceptual processes to test for differences in expectation precision and the expected intensity of pain in the different conditions. This was first assessed in a behavioral study ($n = 54$) that revealed the average order stimulus intensity levels were selected in the conditions with control. This information was then used in an improved task design for a second study that also assessed neural activation using functional magnetic resonance imaging (fMRI) in an independent sample ($n = 59$).

In this work, we show an unpredictability bias in pain ratings, namely that pain of unpredictable intensity is rated as more painful for low intensity and less painful for high intensity stimuli. In addition, we show a similar bias between the controllable and predictable conditions, suggesting an increase in expectation precision and reduced uncertainty with control, a mechanism supported by the results of our modelling analysis and the lower variability in pain ratings. Furthermore, controllability decreases activity in the PAG and supplementary motor area (SMA) during noxious stimulation, while relatively increasing activity in the rostral ACC.

## Results

### Task

Subjects rated individually calibrated heat pain stimuli applied to their forearm using a computerized 0–100 visual analogue scale (VAS). Stimuli had either a low, medium, or high intensity level (30, 50, 70 VAS target). Depending on the condition, trial-wise stimulus intensity was (i) controllable and predictable, (ii) uncontrollable, but predictable, or (iii) uncontrollable and unpredictable (Fig. 1a). We will refer to these conditions as controllable (C), predictable (P), and unpredictable (U). In the controllable condition, participants had to choose a stimulus intensity at the beginning of each trial. In the predictable condition, participants were informed about the upcoming stimulus intensity, which was chosen by the computer program. In the uncontrollable condition, participants remained uninformed about the stimulus intensity, which was again chosen by the computer program. All conditions were matched for motor responses and visual input. The temporal dissociation of motor planning, decision making,

expectation, and outcome phase allowed us to specifically test for differences in brain activity during pain as a function of control. We tested this paradigm in a behavioral sample ($n = 54$) before inviting a sample to undergo fMRI ($n = 59$). After collecting data from the behavioral sample, we were able to match stimulus intensity sequences between the controllable, the predictable, and the unpredictable conditions taking into account systematic effects of choice behavior.

### Control matters: longer reaction times and specific choice patterns in the controllable condition

We observed longer reaction times (RT) in the choice task (controllable condition) compared to the color-matching tasks of the predictable and unpredictable condition (RT in behavioral sample: controllable: $M = 1.76$ s, $SD = 0.84$ s; predictable: $M = 1.46$ s, $SD = 0.87$ s; unpredictable: $M = 1.36$ s, $SD = 0.84$ s; RT in fMRI sample: controllable: $M = 1.72$ s, $SD = 0.66$ s, predictable: $M = 1.09$ s, $SD = 0.47$ s, unpredictable: $M = 1.02$ s, $SD = 0.45$ s; see Supplementary Information, Fig. S6). This underlines that participants made deliberate choices in the controllable condition, which inherently take longer due to another level of cognitive demand.

To check if control systematically influences choice behavior, we analyzed the distribution of chosen intensity frequencies over trials. We observed that participants on average tended to choose high-painful stimuli rather at the beginning, medium-painful stimuli mostly in the middle and low-painful stimuli at the end of a run. To quantify this effect, we investigated the choice data with several linear regression models with Bayesian parameter estimation at all intensity levels. Among the advantages of Bayesian analysis is the ability to generate estimates and credible intervals for the model parameters, such as the slope, in a computationally robust way that does not depend on approximations to hypothetical null distributions, as typical confidence intervals do. In addition, the full probability distributions of the model parameter values, not just point estimates, can be derived from the posterior representing the regression parameters[37].

We report the means of the posteriors of the slope parameters and their 95% interval of highest posterior density (HPDI). HPDIs span the narrowest region containing a specified fraction of the probability mass of the posterior distribution, in this case 95%. Note that in Bayesian inference $p$-values are not applicable, but if the HPDI of a predictor weight (β) with a certain probability mass does not include zero, this provides an indication (at that level of probability) for its influence on the outcome variable. The slopes of linear models were negative for the high intensity level (behavioral sample: $\beta = -3.59$, HPDI = [−4.45, −2.65], fMRI sample: $\beta = -2.63$, HPDI = [−3.42, −1.76]) and positive for the low intensity level (behavioral sample: $\beta = 3.08$, HPDI = [2.18, 3.96], fMRI sample $\beta = 2.21$, HPDI = [1.19, 3.19]). The distribution of choice frequencies of the medium intensity level was better fit by a second order polynomial function than a simple linear model, due to the inverted U-shape present in the data (behavioral sample: $\beta_{lin} = 0.48$, HPDI = [−0.92, 1.92], $\beta_{poly1} = 7.26$, HPDI = [2.25, 12.01], $\beta_{poly2} = -0.42$, HPDI = [−0.71, −0.13], $ELPD_{lin} = -58.9$, $ELPD_{poly} = -55$; fMRI sample: $\beta_{lin} = 0.43$, HPDI = [−0.79, 1.61], $\beta_{poly1} = 6.18$ HPDI = [2.29, 10.06], $\beta_{poly2} = -0.36$, HPDI = [−0.58, −0.11], $ELPD_{lin} = -56.7$, $ELPD_{poly} = -52.9$ (higher ELPD values indicate a better model fit; see Supplementary Information for all parameters estimates and visualization of choice patterns and stimulus intensity sequences). This finding replicates an established effect of negative anticipation of pain, demonstrating that individuals tend to select highly painful stimuli sooner rather than later in order to alleviate dread and overall discomfort[38]. In light of this result, we adjusted the predictable and unpredictable stimulus sequences for the fMRI sample to ensure that the intensity of stimuli across all conditions was matched as closely as possible (see Supplementary Information, Figs. S1–S3).

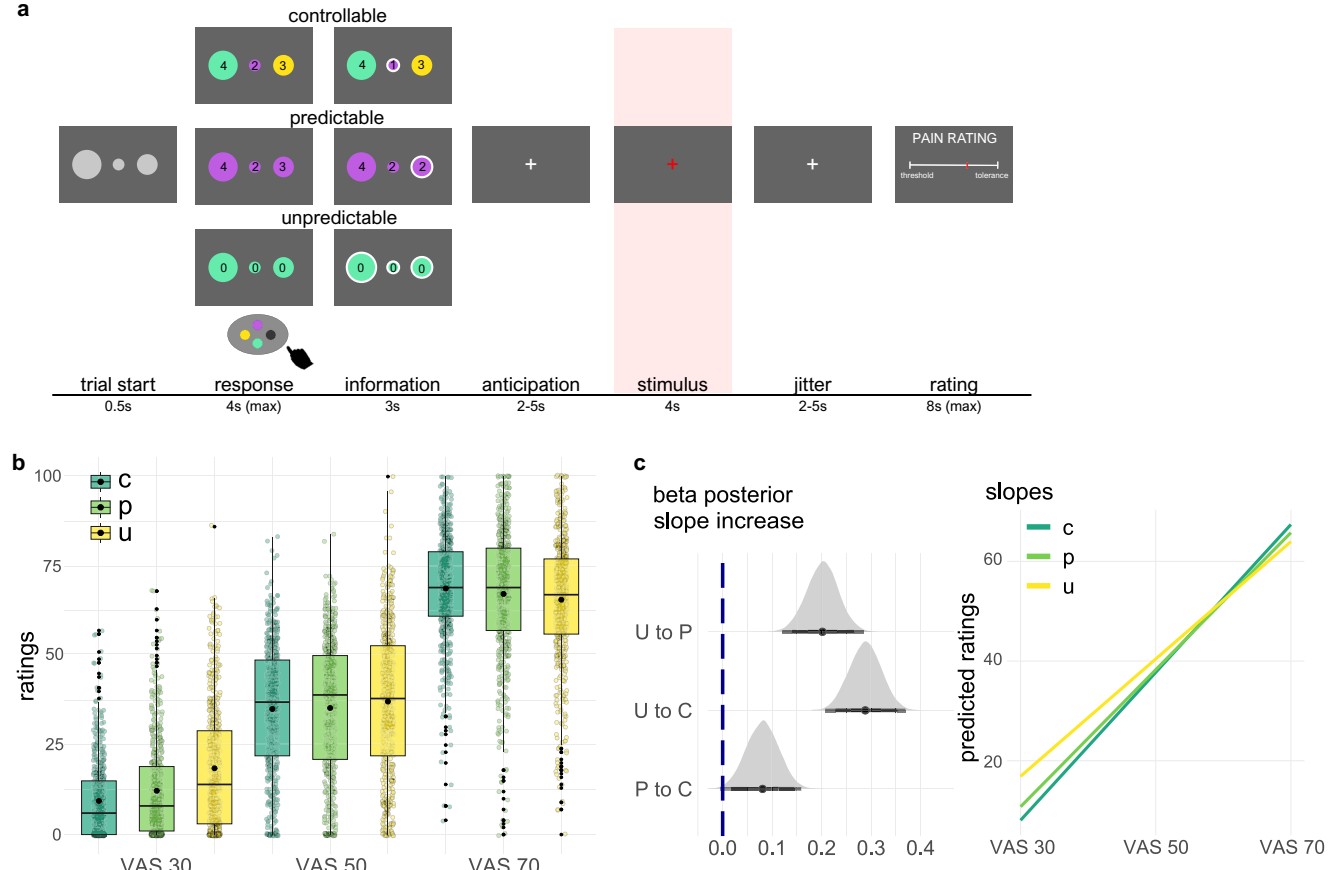

**Fig. 1 | Experimental task, pain ratings, and visualization of interaction effects.**
**a** Time-course of one experimental trial. Participants either chose the upcoming stimulus intensity (controllable runs) or confirmed the color of the circles (predictable, unpredictable runs) by selecting the correct button displayed on the lower part of the screen. Number of remaining stimuli of each intensity level were shown inside the circles in the controllable and predictable condition. In the unpredictable condition a zero was shown inside the circles for the duration of the entire run. To guarantee ratings for all intensity levels in controllable runs, the limitation was imposed that participants had to select each intensity level five times in one run. After the button press, the circle was highlighted, which corresponded to the intensity in size that was either self-chosen (controllable) or script-determined (predictable). In the unpredictable condition, all circles were highlighted for 3 s. After a jittered anticipation phase (2–5 s) that was added to the information period, a white fixation cross turned red and a 4 s long painful heat stimulus of either low (VAS 30) medium (VAS 50), or high (VAS 70) intensity was applied. After stimulus offset the cross turned white again (2–5 s jitter) before the rating scale was shown. Each condition was repeated once, resulting in 6 runs and 90 trials in total. **b** Pain

ratings of the fMRI sample (n = 59 participants; 590 ratings per condition x intensity combination) in the controllable (C, dark green), predictable (P, light green), and unpredictable (U, yellow) condition. Large black points in the boxplots indicate mean values, boxes show the 25th and 75th percentile around the median line, whiskers extend to ± 1.5 × IQR, outliers are shown as small black points. **c** Posterior distribution of interaction parameters and fitted lines from the linear model. The grey curves (posteriors) show the estimated probability densities of the parameter value. A positive value of the interaction parameter (beta) indicates a slope increase over intensities for the indicated condition pair, i.e., when moving from the unpredictable to the predictable (U to P), from the unpredictable to the controllable condition (U to C) and from the predictable to the controllable condition (P to C), ratings show a stronger increase from the low to the high intensity level. Posteriors of the interaction parameters are above zero, implying an effect on ratings. The schematic plot on the right shows the simplified fitted lines derived from model fitting, which illustrate the changes in slope between the controllable, predictable, and unpredictable condition.

## Predictability and controllability effects on pain ratings

To analyze the pain ratings, we applied linear mixed effects models (LMM) with Bayesian parameter estimation. The models implemented the stimulus intensity, the condition identifier, and the trial and session numbers as fixed effects. The subject identifier was included as random intercept effect to account for repeated measurements. In addition to reporting parameter means and HPDI, we refer the reader to the graphical depiction of the posteriors of the predictors of this model in Fig. 1c and Supplementary Information for all predictor estimates. Predictability, controllability, and stimulus intensity interactively altered pain ratings. Ratings were more biased to the extremes of the scale in the predictable than in the unpredictable condition and even more so in the controllable condition. Ratings in the unpredictable condition were biased towards the scale mean. This indicates that ratings were biased towards the expected intensities, when intensity

was predictable (and controllable). Resulting from this effect, participants rated unpredictable low intensity stimuli as more painful than controllable and predictable stimuli of the same intensity (fMRI sample: controllable: $M = 9.39$, $SD = 10.82$; predictable: $M = 12.23$, $SD = 13.46$; unpredictable: $M = 18.51$, $SD = 17.37$; see Fig. 1b; behavioral sample: controllable: $M = 9.83$, $SD = 12.85$; predictable: $M = 10.31$, $SD = 12.47$; unpredictable: $M = 13.22$, SD = 15.45; see Fig. S3a). This effect reversed for the high intensity stimuli, where controllable stimuli were rated as more painful than unpredictable stimuli (fMRI sample: controllable: $M = 68.73$, $SD = 15.12$; predictable: $M = 67.22$, SD = 18.46; unpredictable: $M = 65.60$, $SD = 17.99$; see Fig. 1b; behavioral sample: controllable: $M = 74.58$, $SD = 17.52$; predictable: $M = 73.53$, $SD = 19.27$; unpredictable: $M = 73.93$, $SD = 20.52$; see Fig. S4a). The interaction was most pronounced between the extreme conditions (controllable vs. unpredictable: $\beta_{CxU} = 0.29$, HPDI = [0.23, 0.35], Fig. 1c), but was also

present between the predictable and unpredictable condition ($\beta_{PxU}$ = 0.2, HPDI = [0.14, 0.27], Fig. 1b) and additionally between the controllable and predictable condition in the fMRI sample ($\beta_{CxP}$ = 0.08, HPDI = [0.02, 0.14], Fig. 1b). In the behavioral sample the interaction effects were similar, but smaller, possibly due to the poorer matching of intensity sequences: $\beta_{CxU}$ = 0.07, HPDI = [0.003, 0.13]; $\beta_{PxU}$ = 0.04, HPDI = [−0.02, 0.11]; $\beta_{CxP}$ = 0.02, HPDI = [−0.04, 0.07]; see Supplementary Information, Fig. S4). Because we found that the condition effect crossed over intensity levels, we do not report main effects of conditions here, as this would make the data appear more consistent than they are.

### Predictability and controllability effects on rating variability

The interaction effects indicated differences in the processing of the painful stimuli, possibly due to different levels of expectation precision. This motivated us to take a closer look at the possible mechanisms behind these results. Informed by previous studies[33,39] we hypothesized that these effects could be explained in terms of Bayesian models of pain. In Bayesian models of pain, the perception of pain (posterior), as assessed by pain ratings, results from the integration of the expectation (prior) and the sensory input (likelihood), weighted by their precisions. It follows that the posterior variability is determined by the precision of the likelihood and the prior. However, because we applied calibrated stimuli of the same temperature in all three conditions, the likelihood can be considered constant. Thus, a difference in the variability of ratings (i.e., precision of the posterior) between two conditions can be attributed to a difference in the precision of the prior. Before applying more complex models, we wanted to test, if the standard deviations of pain ratings were indeed different in the controllable, predictable, and unpredictable condition, indicating a change in prior precision. To this end, we applied a new set of LMMs to the within-subject standard deviations of the pain ratings. The models implemented the stimulus intensity, the condition identifier, and the trial and session numbers as fixed effects. The subject identifier was included as random intercept effect to account for repeated measurements.

We observed that pain ratings in the unpredictable condition had the highest within-subject standard deviation, whereas predictability and controllability reduced the standard deviations of the ratings (fMRI sample: $\beta_{UxP}$ = −3.31, HPDI = [−4.33, −2.29]; $\beta_{UxC}$ = −4.41, HPDI = [−5.44, −3.36]; behavioral sample: $\beta_{UxP}$ = −2.09, HPDI = [−3.29, −1.00]; $\beta_{UxC}$ = −2.45, HPDI = [−3.57, −1.30]). Importantly, the standard deviations of ratings were higher in the predictable condition than in the controllable condition in the fMRI sample (controllable: $M$ = 9.71, $SE$ = 0.89; predictable: $M$ = 10.8, $SE$ = 0.99; unpredictable: $M$ = 14.26, $SE$ = 1.31; $\beta_{CxP}$ = 0.98, HPDI = [0.02, 2.04]; Fig. 2a for visualization; behavioral sample: controllable: $M$ = 9.22, $SE$ = 0.89; predictable: $M$ = 9.58, $SE$ = 0.92; unpredictable: $M$ = 11.87, $SE$ = 1.14; see Fig. S5 and Supplementary Information for all predictor estimates). This result is consistent with the hypotheses derived from Bayesian models of pain perception: in the unpredictable condition, the prior is expected to be less precise, resulting in a less precise posterior and more variable pain ratings. In addition, the prior in the unpredictable condition is possibly centered further away from the actual applied intensity (prior mean-shift), which also leads to higher variability in ratings. Importantly, the difference between the controllable and predictable condition point to an additional effect of control on expectation precision.

The standard deviations also differed for the intensity levels (Fig. 2b). Ratings of medium intensity stimuli had a higher standard deviation than ratings of low and high intensity stimuli. The ratings of the high intensity stimuli had higher standard deviations than the ratings of the low intense stimuli (fMRI sample: low: $M$ = 9.39, $SE$ = 0.86; medium: $M$ = 13.52, $SE$ = 1.24; high: $M$ = 11.87, $SD$ = 1.09; behavioral sample: low: $M$ = 7.86, $SE$ = 0.76; medium: $M$ = 12.94, $SE$ = 1.24; high: $M$ = 9.88, $SE$ = 0.95; see Supplementary Information for all predictor estimates).

### Effect of controllability can be better explained by an increase in prior precision than by a mean shift

Even though the cue perfectly predicted the upcoming pain intensity in the controllable and predictable conditions, standard deviations of pain ratings were lower in the controllable condition. One possible explanation for this effect in the context of the Bayesian pain model is that participants form a more precise expectation (i.e., increase in prior precision) when they have control, which consecutively can lead to a difference in posterior means if hard boundaries are present. However, the interactive effects on pain ratings in the controllable condition could in principle also result from a bias in the expected pain intensity in the controllable condition (i.e., a shift in the prior mean). We investigated these competing hypotheses by examining different computational models.

In all models, the prior (expectation) and the likelihood (sensory input) distribution are integrated to form the posterior distribution of pain ratings. The relevant parameters defining the shape of the posterior distribution are the mean and the precision of the distributions. We were interested if additional effects of controllability were due to shifts of prior mean values (mean-shift model) or a change in prior precision (precision-change model). The models can be seen as an approximation to Bayesian integration: under the assumption that the likelihood distribution (i.e., nociception or sensory model component) is identical in both conditions, a change of posterior must be due to changes in the prior distribution. We expected no change in prediction error or likelihood precision in the predictable and controllable conditions and therefore let the model sample directly from a hypothesized prior distribution in the controllable and predictable conditions and examined the different mechanisms, by varying the flexibility of their parameters.

In the mean-shift model, the prior means of the controllable and predictable condition were included as two separate parameters, estimated for each subject, while the precisions of the distributions were constraint to be equal. In the precision-change model, prior means of the controllable and predictable condition were constrained to be equal, but the precisions were left free to vary. In two different versions of both models, the prior mean in the unpredictable condition was defined either (1) as being centered at VAS 50 and fixed throughout all unpredictable trials, or (2) as dynamic, calculated as the summed product of occurrence probabilities and the mean values of stimuli at different intensity levels. This dynamic approach resulted in a value near 50 in most trials, but became more informative towards the end of a run or when many stimuli of the same temperature were applied consecutively. The dynamic model allowed for an update of the unpredictable prior, to reflect, for example, that participants would expect a rather low- or medium-intensity stimulus after having received three high intensity stimuli consecutively. In contrast to the other conditions, we expected the occurrence of prediction errors in the unpredictable trials and therefore could not sample directly from the prior. Thus, for the unpredictable trials, we defined the likelihood mean at the VAS target of the respective trial while accounting for a linear habituation term and an additional parameter (α) to govern the integration process of the likelihood and the prior. A higher value of α indicates a stronger influence of the likelihood on the ratings, whereas a lower value indicates a stronger bias towards the (mean-centered) prior. These models were compared against two null models: a first one included only a habituation term and the second one allowed for different mean values for the controllable and the predictable condition, but did not include the expectation-related bias in the unpredictable condition. Model comparison indicated that the precision-change model explained the data better than the mean-shift model and the two null models (Fig. 3b) supporting our hypothesis derived from the analysis of standard deviations of pain ratings. In addition, both models that included a dynamic, unpredictable prior mean showed better fit to the data, suggesting that participants update the prior

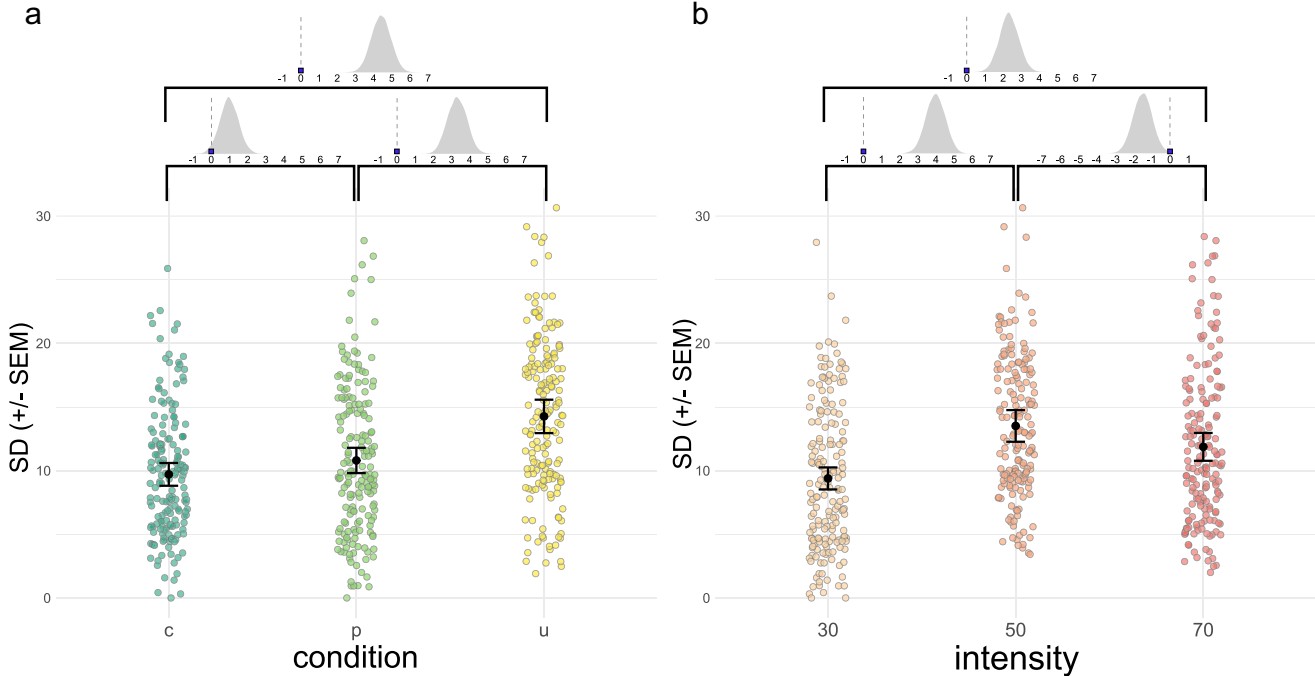

**Fig. 2 | Within-subject standard deviations of pain ratings for conditions and intensity levels. a** Standard deviations of ratings in the controllable condition (dark green points) were lower than in the predictable (light green points) and the unpredictable (yellow points) condition ($n = 59$ values per condition). Black points demarcate the mean standard deviations; error bars the standard error of the mean (SEM) corrected for within-subject measurements. Posteriors of regression parameters (betas) of the pairwise comparisons are displayed above the plots. The grey curves show the estimated probability densities of beta values. If the probability curve is centered at a positive value, this indicates an increase in standard deviation for the condition relative to the reference condition. The value of zero is highlighted on the scale. The posteriors of betas lie above zero for the comparison of the controllable and unpredictable, predictable and unpredictable, and

controllable and predictable condition, indicating a substantial influence. **b** Standard deviations of ratings at the low (beige points) medium (orange points), and the high (red points) intensity level ($n = 59$ values per intensity level). Black points demarcate the mean standard deviations; error bars the standard error of the mean (SEM) corrected for within-subject measurements. Standard deviations of pain ratings were highest at the medium intensity level (VAS 50) and higher for the high intensity level (VAS 70) than low intensity level (VAS 30). The posteriors above the plots show the effect magnitude and distribution; the largest fraction and peak of the probability curve are above zero when moving from the low to the medium or the low to the high intensity level. The curve covers negative values when moving from the medium intensity (as baseline) to the high intensity level.

means across trials. As an aside, the superiority of the expectation-integration models over the null models highlights the significant influence of expectations in the unpredictable condition. As expected, the standard deviation of the prior, estimated by the model, was higher in the predictable than in the controllable condition (see depiction of the dispersion parameter posteriors in Fig. S7). The weighting parameter α was high overall ($M = 0.74$, $SD = 0.18$), due to the overall strong impact of the likelihood (actual calibrated intensity) on ratings in the unpredictable condition.

## Neural bases of predictable and controllable pain processing

To analyze the neural underpinnings of this control-induced effect on expectation precision and pain ratings, we first examined whether heat stimulation produced typical neural response patterns according to the perceived painfulness. To do this, we considered the time when the painful stimulus reached its target temperature plateau (stimulus time window in Fig. 1a) and performed whole-brain analyses with family-wise error correction (WB-corr) for multiple comparisons at $p < 0.05$. We included z-scored pain ratings of the participants as parametric modulators of the onset regressors to identify brain regions that responded to subjectively perceived pain intensity.

This analysis revealed activation in typical pain processing brain regions corresponding to the perceived pain intensity[40,41]. Activity in the central operculum, posterior and anterior insula (AI), somatosensory and motor cortices, middle cingulate cortex, parietal cortex, cerebellum, and the brainstem were positively related to the pain ratings (all $p_{WB\text{-}corr} < 0.05$, max. $T = 12.76$, height threshold $T = 5.32$, DF = [1.0, 162]; see Fig. 4).

## Predictability and controllability affect brain activity during pain

We next investigated the neural processes engaged by unpredictable, predictable computer-chosen, and controllable (predictable self-chosen) pain. We first computed a contrast between the two predictable and the unpredictable condition (C, P vs. U), and in a second step contrasted the predictable and controllable condition, to detect any additional effects of control on processing of the painful stimulus (C vs. P) in both directions.

During pain of predictable intensity, we observed decreased activity in parietal, frontal and (pre)motor regions, including the anterior insula, the SMA, middle cingulate cortex, ACC, cerebellum and PAG compared to the unpredictable conditions, as shown in Fig. 5a (contrast: C, P < U; $p_{WB\text{-}corr} < 0.05$; max. $T = 10.49$, height threshold $T = 5.31$, DF = [1.0, 162.0]).

Importantly, comparing the controllable to the predictable condition (C < P), we found that activity was additionally decreased in the SMA, dorsal ACC, PAG, precuneus, parietal cortex and cerebellum ($p_{WB\text{-}corr} < 0.05$; max. $T = 6.72$, height threshold $T = 5.31$, DF = [1.0, 162.0]) (Fig. 5b). In most of the regions we found a parametric increase in activity from controllable to predictable and to unpredictable stimulation, suggesting a relationship with increasing expectation uncertainty.

The reverse contrast (C, P > U) revealed increased activity in rostral parts of the ACC (rACC; Fig. 5c), hippocampus and precuneus ($p_{WB\text{-}corr} < 0.05$; max. $T = 6.02$, height threshold $T = 5.31$, DF = [1.0, 162.0]). The rACC and the central operculum responded additionally to controllability ($p_{WB\text{-}corr} < 0.05$; max. $T = 5.53$, height threshold $T = 5.31$, DF = [1.0, 162.0]) (Fig. 5d). Extracted parameter estimates from ACC

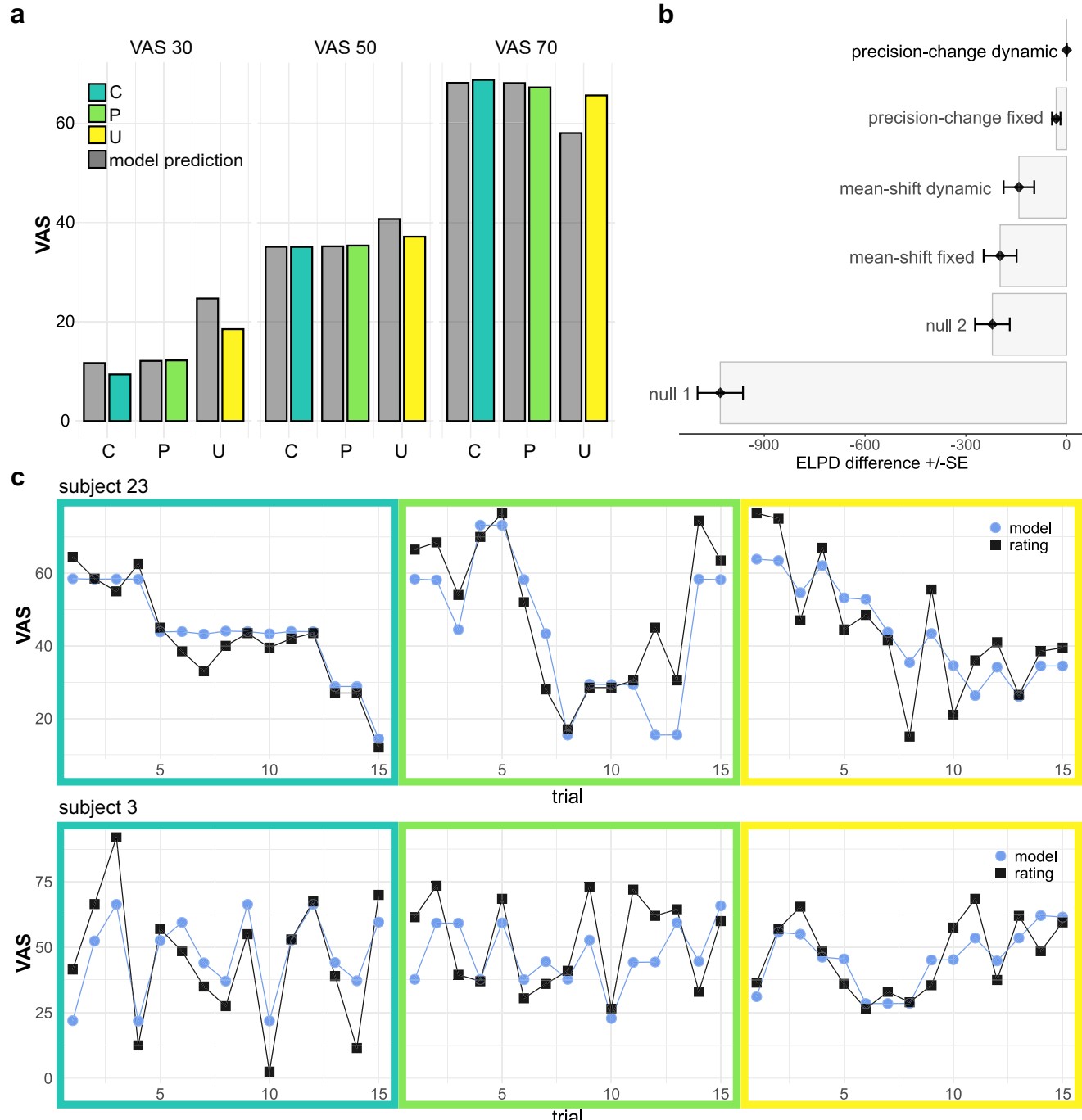

**Fig. 3 | Model predictions and comparison. a** Schematic group-level predictions of the winning model (in grey) follow closely the empirical average pain ratings (green to yellow). **b** Model comparison using the robust version of leave-one-out cross validation. Expected log point-wise predictive density (ELPD) was compared between models. Higher ELPD values indicate better model fit. The winning precision-change model implemented a dynamic prior mean in the unpredictable condition and shared posterior means for the controllable and predictable condition. Precisions were allowed to vary between the conditions. **c** Model predictions for pain ratings in two exemplary single subjects. Black squares indicate actual ratings and blue circles indicate model predictions.

regions show a parametric decrease in activity from controllable to predictable and unpredictable painful stimulation, while the difference between controllable and predictable trials was especially pronounced in pregenual ACC, close to vmPFC (Fig. 5d).

**Interaction effect of condition and intensity level on brain activity during pain**

We also explored if an interaction effect of condition and intensity level would be visible in brain activity. At the whole brain FWE corrected significance threshold (at $p < 0.05$), no voxel was significant for any of the interaction contrasts. Post-hoc visual inspection of the interaction effects at an uncorrected threshold of $p < 0.001$ revealed an activation cluster in the rostral ACC, showing the same pattern as the pain ratings, i.e., activity in this ACC cluster positively scaled with intensity in the controllable condition and negatively with intensity in the unpredictable condition (see Supplementary Information, Fig. S14). However, this effect is clearly established post-hoc and would only be significant if using an ACC mask and performing small volume (SV) correction ($p_{SV\text{-}corr.} = 0.003$ in voxel = [−7.5, 37.5, −1.5]; $T = 4.96$).

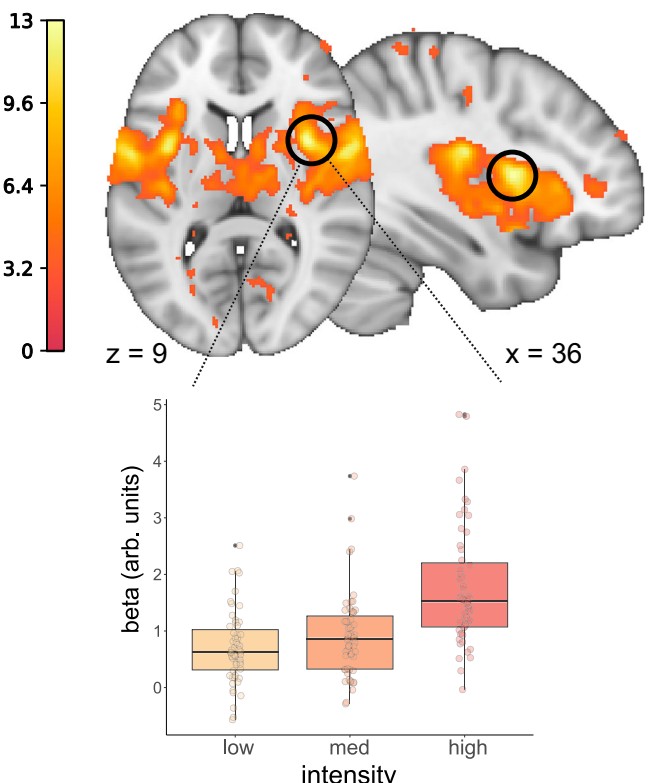

**Fig. 4 | Main effect of pain.** Contrast image displays parametric effects of perceived painfulness. Stimulus onsets were modulated with z-scored pain ratings. The region with highest increase related to perceived painfulness is the central operculum. For display purposes, we extracted the parameter estimates from a model with separate regressors for low, medium and highly painful stimulation. Box plots show the estimates derived from the peak voxel of the parametric contrast (central operculum: [36, 8, 9]). Boxes show the 25th and 75th percentile around the median line, whiskers extend to ±1.5 × IQR, outliers are shown as small black points. Points show the beta estimates for each individual subject. The estimates result from first-level analyses of $n = 55$ participants.

## Discussion

This within-subject study investigated the mechanisms of how control can modulate aversive experiences, i.e., pain. We compared behavioral and neural processes elicited by stimuli of either controllable, predictable, or unpredictable intensity. Including a predictable condition provided an important control condition. With respect to pain ratings, our data show a bias towards the predicted values in the predictable and controllable condition, where the upcoming stimulus intensity was precisely known and a bias towards the mean intensity in the unpredictable condition, where stimulus intensity could not be predicted. This expectation precision effect was even stronger for the controllable condition, resulting in an interaction of pain predictability, controllability, and intensity. The variability of pain ratings, which serve as a proxy for prior uncertainty, were higher in the predictable condition compared to the controllable condition. Computational modelling indicated that an increase in expectation precision, rather than shifts in expectations, was more likely to explain the results. With respect to neuronal activity, the rostral anterior cingulate cortex (rACC), the PAG, and the SMA show effects associated with the changes in expectation precision, when comparing controllable and predictable stimuli to unpredictable stimuli. In addition, these brain regions also exhibit clear additional signal differences for controllable stimuli.

Motivated by findings concerning learned helplessness in animals, control over aversive stimuli was thought to recruit specific safety-signaling pathways[4]. These studies usually employed controllability

manipulations that were often confounded by outcome predictability, i.e., without control, the uncertainty regarding features of the aversive stimulus was also increased[22,42]. The uncertainty of outcomes has already been discussed as a possible explanation for a perceived loss of control in early studies where unpredictability was found to confound investigations of control over stress[42,43], but only recently led to adaptations in task design and formalization of control to actually measure perceived control over state transitions separated from predictability[2,44]. Importantly, only the features of the stimulus that are under control become more predictable and inform perceived certainty. For instance, if the offset of a stimulus can be controlled, this does not lead to increased certainty about its intensity. Conversely, control over stimulus intensity does possibly not change the estimate of its occurrence probability. So, the general feeling of uncertainty relies on more than one aspect of the stimulus and individual and possible context-dependent importance weighting of different features might also be relevant in that regard.

Still, our data suggests that control actually induces an increase in outcome certainty, more than mere predictability, which leads to modulations in the perception of aversive stimuli. This resulted in (i) an interaction effect across intensity levels and (ii) lower standard deviations of within-subject ratings in the controllable condition, in which stimulus intensity could be self-determined, compared to a condition in which the intensity was perfectly predictable but not self-chosen (i.e., controllable).

The interaction effects over intensities replicate former findings on predictability modulations of perception and can be explained by Bayesian integration processes of expectation and sensory input[27,28,45,46]. When the intensity of a stimulus can be correctly predicted, the expectation about its painfulness is centered on the predicted intensity, which will match the actual stimulation. In contrast, when expectations are uncertain, like in the unpredictable condition, the best guess about the upcoming stimulus intensity is the mean of all possible stimulus intensities. This leads to a bias towards the mean intensity in conditions entailing lower outcome certainty, which is also present in our data. Surprisingly, we also observed this effect when contrasting the controllable with the predictable condition. All stimuli were pre-calibrated to subjective pain perception and kept constant throughout all conditions, thus, the sensory input was identical between conditions. Therefore, changes in ratings must stem from changes in expectations (e.g., the prior distribution) when thinking in terms of expectation-sensation integration models.

This is interesting as a recent study reported that agency over pain treatment lead to a shift in expected painfulness (i.e., a mean shift of the prior)[26]. Consequently, we tested whether different prior means could also have produced the difference between the controllable and predictable condition (mean-shift) in our data. However, the mean-shift model performed worse in explaining the data than a model, which allowed changes in expectation precision. Therefore, we conclude that differences in expectation precision and not mean values led to the effect of control on pain perception. This also explains why pain ratings were less variable in the controllable condition. Note that the interaction effect on ratings can only result from prior precision changes (and not prior mean changes) in scenarios where there are hard boundaries on the perceptual posterior distributions, like the pain threshold and pain tolerance in our case.

Our study investigated control over pain intensity. However, control has a broad dimensionality and can also be exerted, for example, over the onset, duration, or location of a stimulus[47]. We believe that the increase in expectation precision could be a significant factor for all types of control, with respect to the feature of the stimulus over which control is exerted. Other types of control might involve additional mechanisms, e.g., when self-touch or motion are part of the control process[26,48].

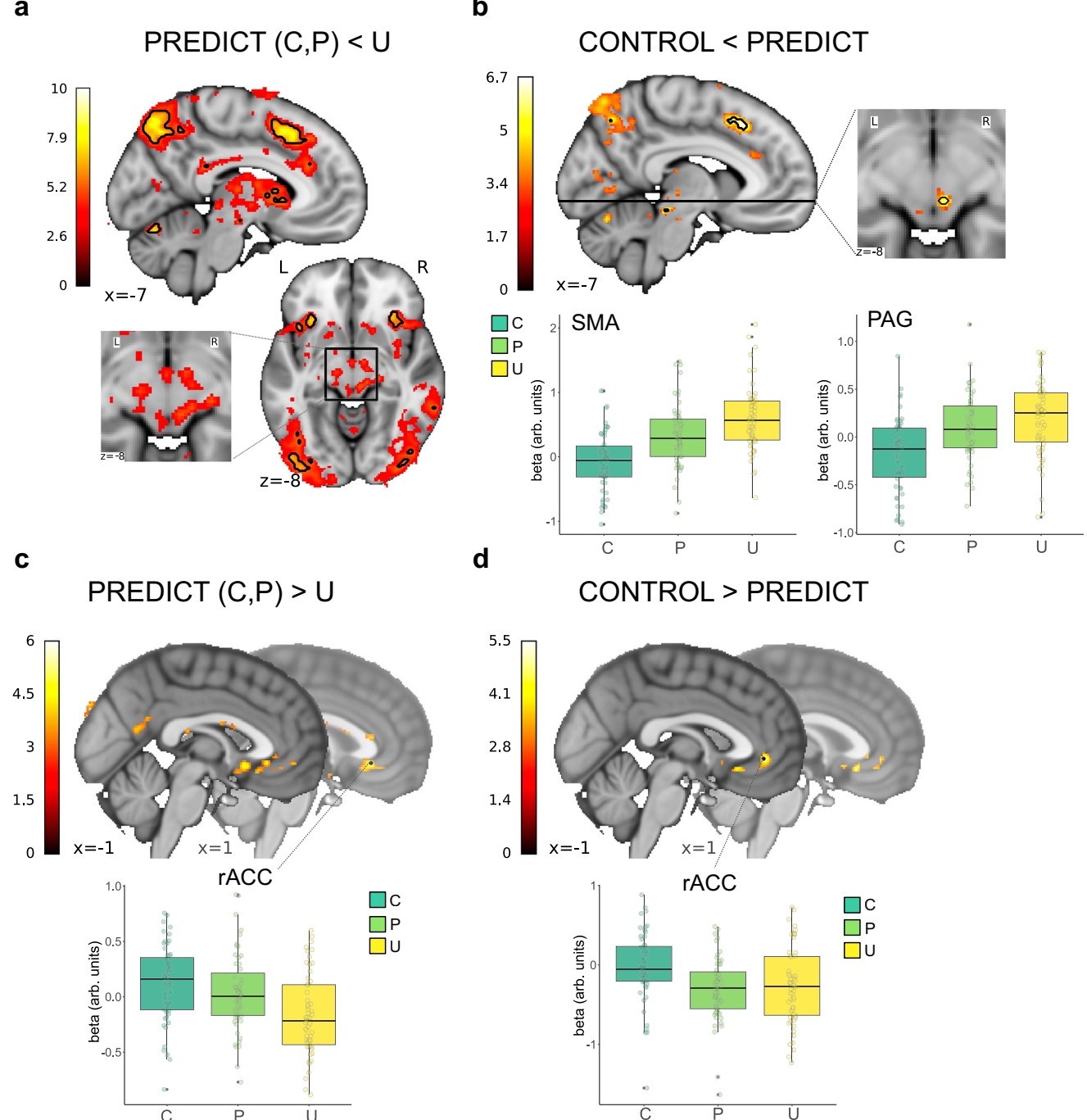

**Fig. 5 | Effects of predictability and controllability on brain activity during pain.** Contrasts are derived from statistical maps at $p < 0.001$, for better visibility, a threshold of $T = 3.0$ was applied for the contours. All contrasts are based on one-way ANOVAs with correction for multiple comparisons on whole-brain level. For boxplots, parameter estimates were derived from voxels that were significant at $p < 0.05$ corrected for multiple comparisons on whole-brain level. **a** Contrast image depicting regions of increased activity in response to unpredictable pain (C, P < U) in hot colors. Black outlines indicate regions that were significant at $p < 0.05$ after correction for multiple comparisons. **b** Contrast image showing regions in which activity was additionally reduced in the controllable condition (C < P) with contour lines for $p_{\text{WB-corr}} < 0.05$. The parameter estimates in the three conditions from the SMA [−7.5, 21, 45] and PAG [6, −27, −7.5] are shown below. Both regions exhibited an effect of predictability and controllability. Boxplots show beta estimates for the peak voxel of the contrast C < P. Boxes show the 25th and 75th percentile around the median line, whiskers extend to ± 1.5 × IQR, outliers are shown as small black

points. Points show the beta estimates for each individual subject. The estimates result from first-level analyses of $n = 55$ participants. **c** Contrast image depicting regions of higher activity during controllable and predictable pain. The BOLD signal was higher during predictable pain in the rACC [1.5, 28.5, −6]. Boxplots show beta estimates for the peak voxel of the contrast C, P > U. Boxes show the 25th and 75th percentile around the median line, whiskers extend to ± 1.5 × IQR, outliers are shown as small black points. Points show the beta estimates for each individual subject. The estimates result from first-level analyses of $n = 55$ participants. **d** A more frontal part of the rACC responded specifically to controllability [−1.5, 40, −3]. Boxplots show beta estimates for the peak voxel of the contrast C > P. Boxes show the 25th and 75th percentile around the median line, whiskers extend to ± 1.5 × IQR, outliers are shown as small black points. Points show the beta estimates for each individual subject. The estimates result from first-level analyses of $n = 55$ participants.

Also, different results can emerge when control is exercised over the initiation of aversive stimuli in contrast to positive events. The increase in expectation precision might amplify the perceived effectiveness of a treatment when it is self-controlled[49], whereas it may result in higher pain ratings when control is exercised over the onset of a nocebo intervention[29]. Following the Bayesian model framework, this could then also be explained as a unidirectional prior mean-shift, as has been reported in a study investigating self-treatment[26].

The different levels of expectation uncertainty also led to changes in brain activity during pain. Unpredictable stimulus intensities led to increased activity in the anterior insula, dACC, parietal, middle frontal gyrus, and the cerebellum, regions typically recruited in response to unexpected and potentially behaviorally relevant stimuli, for example, pain of unknown intensity[36,41]. Crucially, activity in the (pre-)SMA and PAG was attenuated not only by predictability, but was also significantly lower when the stimulus intensity was self-chosen by the participants. Activity in rACC on the other hand was increased in response to the controllability manipulation.

The SMA is a critical region for motor control, monitoring signals that might be relevant to a change in motor behavior and was shown to be implicated in movement execution and inhibition[50], for example, SMA neurons encode covert errors in movements and thus enable last minute inhibition of an erroneous response[51].

Pain is a highly informative signal that indicates the need for behavioral change and control of motor output, e.g., contact with an increasingly noxious hot plate requires ongoing assessment of the painfulness induced and planning and control of the movement required to move the body away from the stimulus. It has experimentally been shown, that activity in the SMA and neighboring cingulate motor areas is related to the modulation of pain-related motor responses[52,53] and evidence accumulation regarding painfulness[54]. SMA activity also increases when a painful stimulus of unexpected intensity is applied[55,56]. This is in agreement with our data, showing that SMA activity not only decreased with predictability, but was additionally attenuated during pain of self-chosen intensities. This further suggests that due to higher expectation precision, the demand to monitor stimulus intensity and prepare pain-related motor responses was reduced in the controllable condition. Uncertainty about the painfulness on the other hand, increased the need for evidence accumulation and motor preparation in the two non-controllable conditions, leading to more activity in the SMA.

The observed activation covered the SMA and pre-SMA. While the SMA mainly responds to pain and connects to motor regions, the pre-SMA is linked to higher-order cognitive systems[57,58]. Some accounts suggest a functional gradient from allocating motor control (i.e., in a Stroop task) to higher-order cognitive control (i.e a difficult decision) from the SMA through the pre-SMA to the dACC[59,60]. The activation extending to the pre-SMA in our study suggests that control over pain not only affects motor preparation, but also higher-order attention allocation, which may be necessary for pain evaluation.

Another region that encoded the difference between the controllable and predictable condition was the PAG, where activity was reduced in the controllable condition. The PAG plays a crucial role in pain modulation and has been associated with placebo hypoalgesia[61,62], but also with hyperalgesic effects of uncertainty[34] and nocebo treatments[32]. The PAG plays a bi-directional role in pain modulation as it can contribute to either pain inhibition of facilitation[32], possibly via differential activation of on-or-off cell in the RVM[63].

The PAG is also responsive to differences in expectation precision: a study that explicitly manipulated prior precision showed that PAG activity was lowered during pain, when more precise information about its intensity was available[33]. Further studies reported increased PAG activity when the intensity of a painful heat stimulus was unpredictable[64,65]. These findings are corroborated by animal studies, indicating that the PAG responds to unexpected shocks during fear learning, but stops to encode aversive stimuli after conditioning, i.e., when the outcome is certain[66,67]. The PAG has also been shown to respond to pain-related prediction errors[68], indicating its pivotal role in integrating expectations with ascending nociceptive information[69] and reflecting the outcome of the Bayesian integration process[33].

The PAG receives inputs from rACC[35], that also responded parametrically to the predictability and controllability manipulation in our study. The rACC is part of the opiodergic descending pain modulatory system[61,70], involved in the effect of opioid analgesia[71], is recruited during distraction-induced hypoalgesia[72,73], and mediates expectation effects on pain in placebo studies[30,61,74]. These results were recently supported by a rodent study showing that rACC neurons are crucially involved in pain modulation by expectations of pain relief[75] and by a study in a human sample showing that rACC-PAG coupling increased in response to boosted placebo effects through side effects[31]. Interestingly, this study and a previous study[70] partially reported decreased BOLD signals in the rACC during the active placebo condition suggesting that the rACC has neither purely anti-nociceptive nor pronociceptive properties

Our paradigm consisted of a task-stimulus sequence (i.e., decision task followed by pain). This raises the question of whether the task evoked BOLD effect could influence the pain related activation in analogy to a repetition suppression effect[76,77]. Although this is unlikely given the rather long interval between task and pain onset and an additional jitter, we further investigated this using a time-resolved fMRI analysis (i.e., using a FIR model). Even though this analysis showed that the BOLD signal in the SMA and PAG was initially higher in the choice compared to the color-matching task, activity levels at pain onset were not significantly different between the conditions (see Supplementary Information, Fig. S15).

In summary, the fMRI results are supportive and consistent with the behavioral results. Apart from the interaction effect present in the rACC, the increase in BOLD signal in salience and attention processing brain areas, including the SMA and the PAG, reflect the relatively lower expectation precision in the unpredictable and predictable condition relative to the controllable condition. While we found an additional effect of controllability compared to predictability, our analysis also shows that there is a considerable overlap in effects on behavior and neural activity between both environmental features. This highlights the need for predictability-matched control conditions when investigating control-induced pain modulation. The fMRI results fit the mechanism that was also put forward by our modeling analysis of behavior. Nevertheless, the exact mapping between changes in BOLD signal and psychological effects is often not known and it is therefore difficult to establish a direct causal relationship between both levels of analysis.

We could show that behavioral effects and brain activity changes that are usually reported to result from control over pain can partly be attributed to pain predictability. However, our data also shows a recursive effect of control on expectation precision, which is increased by control and thereby modulates pain. On top of mere predictability, controllability induced uncertainty reduction, which led to activity changes in brain regions that accumulate evidence for the adaptation of motor plans under uncertainty (SMA), mediate pain related expectation effects (rACC), and integrate expectation with sensory input under uncertainty during pain (PAG). These results highlight the important influence of expectation effects on processing of aversive stimuli that are clinically relevant for the effects of helplessness and stress in chronic pain. Furthermore, the increase of expectation precision by control bears important implications for understanding the role of self-management of pain. Our findings suggest that lack of control, as experienced in chronic pain, leads to reduced precision of expectations of positive treatment outcomes and thus may reduce treatment effectiveness through a reduced subjective sense of control.

## Methods

### Sample

Participants were recruited via a local online platform. Screening for exclusion criteria was completed via e-mail and phone. All participants provided informed consent and were paid for participation. The local ethics committee (Ethikkommission der Ärztekammer Hamburg) approved the study. Participants in the fMRI sample additionally underwent an MRI screening with a medical doctor.

Fifty-nine participants took part in the behavioral pilot study (mean age = 26.8, range 19 to 39, 25 male). Participants were included if they met none of the exclusion criteria: acute or chronic somatic or psychiatric disease, drug or medication intake (except oral contraception, allergy & thyroid medication), chronic or acute pain condition. Five participants had to be excluded from the analysis (four made to many errors in the color-matching task (>10% of trials), one person did not understand the task). Remaining data from 54 participants were analyzed.

For the main study, we were aiming for a similar sample size as in the pilot study, plus potential drop outs. We performed an additional power analysis, where we simulated data under different sample sizes and applied linear mixed models. Based on this analysis, we aimed for $N = 60$ participants.

Another sample of 64 participants took part in the fMRI study (mean age = 26.26, range 18–40; 28 male). In addition to the exclusion criteria above participants were screened for MRI contraindications (e.g., pregnancy, metal implants, claustrophobia). Five participants had to be excluded from the analysis (two participants had a pain threshold that was too low to guarantee nociceptive processing, one person thought that they were deceived by the experimenter, one did not understand the task and made too many errors (>10% of trials) in the color-matching task, one person had an incidental MR finding). A final behavioral data set of 59 subjects was analyzed from this sample. MRI data of four subjects had to be excluded from the analysis due to excessive or repeated head movement upon visual inspection, resulting in a total of 55 individual data sets for the fMRI analysis.

### Thermal stimulation

Thermal stimulation was performed using a $30 \times 30$ mm$^2$ Peltier element Thermode (Pathway model in the behavioral lab, TSA 2 in the MR scanner with CHEPS probe, all made by Medoc, Israel). The baseline temperature was set to 32 °C. Temperature rise rate was set to 15 °C/s (behavioral sample) and 13 °C/s (fMRI sample). The difference in rise rates was due to different software limitations for the Pathway and the TSA 2 model at the time of data collection. Throughout the experiment, individually calibrated temperatures were applied for a 4 s long plateau always decreasing to the baseline between stimuli. For safety reasons, we set the maximally allowed temperature to 49 °C. Applied intensities corresponded to low, medium, and high pain intensity (ratings of 30, 50, and 70 on a VAS ranging from 0 to 100). The thermode was placed on three different locations on the participants' left forearm. To limit habituation and sensitization effects the location was changed after each run. The positioning was pseudo-randomized to guarantee the same amount of recovery time for each skin patch and to counterbalance conditions (exemplary randomization schedule see Fig. S6).

### Calibration

Prior to the main task, individual pain perception was determined in a calibration procedure. Depending on the rating of a first pre-exposure stimulus, participants provided binary ratings for increasingly painful stimuli as being painful or non-painful in order to identify individual pain thresholds. Following threshold determination, stimuli of varying intensities were applied to cover the whole range from individual pain threshold to pain tolerance level. Individual linear regression parameters were estimated and validated in an iterative fashion to derive the link between VAS ratings and temperature. In the behavioral lab,

the procedure stopped after obtaining ratings on all 20-point bins of the VAS or after a maximum of 12 trials. The fMRI sample completed this procedure while being in the MR scanner to get acquainted to the scanner environment. To limit effects of habituation and to keep the procedure short in the scanner environment, a maximum of 6 trials was set for the regression validation. The calibration resulted in average temperatures (mean ± standard deviation) of 43.56 °C ± 2.63 (VAS 30), 45.31 °C ± 1.66 (VAS 50), and 47.07 °C ± 1.14 (VAS 70) in the behavioral sample and 44.31 °C ± 2.06 (VAS 30), 45.79 °C ± 1.67 (VAS 50), and 47.2 °C ± 1.39 (VAS 70) in the fMRI sample (see Supplementary Information). The procedure took approximately 15 min. in the behavioral lab and 10 min. in the MRI scanner.

### Task

To investigate the differential influence of controllability and predictability on pain we designed a task with three different conditions, in which the intensity of painful stimuli was either controllable, predictable, or uncontrollable and unpredictable.

In the controllable condition, participants could choose the upcoming stimulus intensity among three different intensity levels (low, medium, high), depicted by three differently sized (small, medium, large) and colored (yellow, mint, violet) circles. The size of each circle corresponded to an intensity level. Choice was executed by selecting a key on a keyboard (behavioral lab) or button on a response box (MR scanner) that matched the circle of the desired size in color (e.g., if participants wanted to select the low intensity, they would have to press the button of the same color as the smallest circle). Participants had 4 s to make and enter their decision. Then, the circle corresponding to the chosen intensity was highlighted and after a jittered expectation phase of 2–5 s the stimulus was applied. Stimulus intensity always matched the choice of the participants, so the task involved no deception. Participants' choices were limited in a way that each intensity had to be selected 5 times in one run of 15 trials. Numbers displayed in the middle of the circles indicated how many stimuli of each intensity were left. After choosing an intensity for the fifth time, a zero was displayed in the middle of the corresponding circle. If participants tried to choose an intensity more than five times, one of the remaining intensities would be randomly chosen by the computer. Participants were instructed to avoid this behavior but to select deliberately. The position and color mapping of the circles and buttons was randomized across trials to control for perceptual similarity and spatial remapping. After stimulus offset and a jittered fixation cross (2–5 s) participants performed the pain rating by moving a red bar on a rating scale displayed on the screen with left and right keys. Participants had to complete their rating within eight seconds. The starting position of the bar was randomized between the value of 30 and 70 on the scale to avoid systematic anchoring effects. After the 8 s rating period, a white fixation cross was shown for a 2–4 s long inter-trial-interval (ITI) before the next trial started.

In the predictable and unpredictable condition, the trials started also by showing three differently sized circles, but in contrast to the controllable condition, all circles were displayed in the same color. The task for the participants was to select the button that had the same color as the circles shown on the screen (*color-matching-task*). We included this task to control for button presses and cognitive demands in the MR-experiment. Differently to the controllable condition, the button press had no influence on the intensity of the following painful stimulus. In the predictable (non-controllable) condition, the numbers of the remaining stimuli of each intensity level were shown inside the circles throughout the run. After the color-matching response participants were informed about the upcoming stimulus intensity through highlighting the circle of the corresponding size. In the unpredictable condition, participants were neither informed about the upcoming stimulus intensity in the current trial (all circles were highlighted instead of one) nor about the number of remaining stimuli of each

intensity level (zeros were displayed inside all circles from beginning until the end of a run). The intensity sequences in the predictable and unpredictable runs were predefined by the computer program (see next paragraph for a detailed description).

Each condition was repeated once, resulting in 6 runs in total, and the condition ordering was pseudo-randomized to make sure that each condition appeared once in the first and in the second half of the experiment. Each run consisted of 15 trials (resulting in 30 trials for each condition and 90 trials in total) and lasted around 7 min. Before the first run of each condition, participants had to perform three test trials of the condition. These test trials could be repeated until participants understood the task. The Investigators were not blinded to the conditions during the experiment and statistical analyses.

### Intensity sequence generation and matching

In the behavioral sample, stimulus intensity sequences in the predictable and unpredictable conditions were free to vary completely at random. This resulted in equally likely occurrence of each intensity level in all trials of predictable or unpredictable runs. However, when we analyzed choice behavior in the controllable condition, we detected that our participants selected highly painful stimuli rather at the beginning than at the end of a run. After observing this effect, we decided to adapt the intensity sequence generation for the predictable and unpredictable conditions in the fMRI sample to make intensity sequences comparable across conditions. This was done by computing the percentage of the chosen stimulus intensities in all trials. These percentages were then used as probabilities for selecting each intensity level in our newly adapted stimulus sequence. Due to the probabilistic structure and the limit of 5 trials per intensity in the sequence generating function, we observed an unwanted increase of high intensity stimuli towards the end of the runs (see Fig. S2). We then adapted the intensity sequence generation method structure again, approximately after half of the data collection of the fMRI sample was finished ($n = 34$ in group 1, $n = 25$ in group 2 that entered the final analysis). To get better-matched intensity sequences for the predictable and unpredictable condition of the second half of the fMRI sample, we simply used stimulus sequences of controllable conditions obtained from participants in the behavioral pilot study in order to make the conditions as comparable as possible.

### Behavioral data analysis

Behavioral data were analyzed using MATLAB R2020a[78], R(v4.0.2)[79], and Stan[80]. We analyzed differences in choice behavior, pain intensity ratings, and within-subject standard deviations of ratings. We analyzed the data of the behavioral and the fMRI sample separately. To check for robustness of our Bayesian approach, we analyzed the data also with a conventional approach to implement linear mixed models in R, using lme4[81] (see Supplementary Information). The fMRI sample was first analyzed taking into account the different ways of intensity sequencing as two different groups. We compared data patterns in the two subsamples: the samples differed in effect sizes, but the triple interactions of intensity, condition, and sample half (demarcating the first and second approach to set up the sequences) were not significant (Supplementary Information), so we decided to finally analyze the fMRI sample as a whole. Finally, we also analyzed the combined data set ($N = 113$, including all participants of the behavioral, fMRI first and second half) to check the effects for robustness, which resulted in the same significant effects as reported for the fMRI sample. As predictors, the models included the trial and session number, the condition and intensity interaction and the subject identifier as random intercept effect.

### Linear models

For our main analysis, data were analyzed using linear models in RStan[80]. Instead of least squares estimation, model parameters (predictor weights, intercepts, error terms) were estimated using the *No-U-*

*Turn sampler* (NUTS), an adaptive type of a Hamiltonian Monte Carlo sampler implemented in Stan[80,82]. Instead of minimizing residuals via the least squares method, NUTS determines the most likely values of model parameters by running physical simulations and obtaining samples of parameter values from the posterior distribution. As in other models, the structure of the random and fixed factors is provided by a design matrix. As a result, probability density distributions of predictor weights can be obtained from the posterior samples for each parameter. The implementation of the linear models followed the general structure of a regression model in Stan (https://mc-stan.org/docs/stan-users-guide/regression.html). Model fitting diagnostics (chain convergence, pairs plots) were examined visually for all models and are summarized in the code repository.

In the simple linear model with fixed effects (only to analyze choice frequency), trial number served as independent and the percentage of chosen stimuli as the dependent variable. Separate models were fitted for each intensity level and for both samples. We fitted an intercept, a slope, and an error term. To account for the inverted U-shape in the medium intensity, we fitted a model with an additional quadratic term. Priors were flat and uniformly distributed over ($-\infty$, $+\infty$) for the models on choice frequency.

Pain intensity ratings were analyzed using LMMs with the predictors of interest (condition, intensity, interactions condition × intensity). Session number and trial-in-session number were also included to account for habituation/sensitization effects. The subject identifier was included as random intercept effect to account for repeated measurements. Two sets of LMMs were fitted on the rating data of each study sample. In the first LMM, the controllable condition was defined as baseline and in the second LMM the predictable condition served as baseline to be able to provide all possible pairwise interactions, in both models the pain threshold (VAS = 0) was set as the reference for the linear intensity predictor. The reference or baseline category in linear models indicates to what average value the influence of the predictors is compared to.

To gain insight in the variability of ratings in the different conditions, we also analyzed within-subject standard deviations of ratings for all three conditions and intensities. Variability in the ratings can be seen as a proxy for expectation uncertainty, because the actual physical input was kept constant in all three conditions. The design matrix differed slightly, because we included intensity as a factor with three distinct levels (low, medium, high) in the analysis, and not as a linear term (30 to 70) because we observed an inverted U-shape pattern in the data upon visual inspection of the raw data. This resulted also in a slightly different interaction pattern, as all factor levels had to be compared to one another (pairwise differences between intensity levels: high vs. med, low vs. high, low vs. med; differences between conditions: P vs. C, U vs. C, U vs. P). The model was fit with two different baseline settings to derive all pairwise comparisons: once with the uncontrollable condition and low stimulus intensity and then with the controllable condition and the medium intensity as baseline levels.

For the mixed effects model analysis, we used the approach outlined in Sorensen et al. (2016)[83]; here, additional random intercepts were estimated for each subject to account for correlated error structure due to repeated measurements. To quantify the robustness of the effects of all model parameters, we report highest posterior density intervals (HDPIs). HPDIs contain a defined amount of the probability mass, e.g., 95%, of the posteriors of the predictor weights. If most of the probability mass of the estimated parameter value does not cover zero it is appropriate to claim that the predictor has an influence on the ratings. We used the following sampling settings for all mixed models: number of iterations = 4000, number of chains = 4, number of warm up iterations = 1000. We set weakly informative regularizing priors for the intercept ($\alpha \sim N(0, 50)$) and the predictor weights ($\beta \sim cauchy(0, 2.5)$) the error term was constrained to be positive and had a uniform prior[82,84,85].

## Bayesian expectation-integration models

To further investigate the influence of expectations and control on pain across conditions, we aimed to fit more theoretically informed models to the rating data.

Prior investigations suggest that pain perception follows Bayesian expectation-integration mechanisms[23–26]; thus, we hypothesized that participants' pain ratings would result from the integration of expectations (prior) with perceived sensory input (likelihood). To establish a basic model, we formalized this process as the integration of two Gaussians, with pain ratings treated as draws from the resulting posterior distribution[26,33].

As by the analytical solution of a normal-normal integration, the posterior mean $\mu_1$ is the precision-weighted and normalized sum of the prior mean $\mu_0$ and likelihood mean $y$. The posterior precision is the sum of the prior precision $\frac{1}{\tau_0^2}$ and likelihood precision $\frac{1}{\sigma^2}$. Thus, the posterior distribution of the pain ratings $\theta$ can be written as $\theta \sim N(\mu_1, \tau_1^2)$ with:

$$\mu_1 = \frac{\frac{1}{\tau_0^2}\mu_0 + \frac{1}{\sigma^2}y}{\frac{1}{\tau_0^2} + \frac{1}{\sigma^2}} \quad (1)$$

$$\frac{1}{\tau_1^2} = \frac{1}{\tau_0^2} + \frac{1}{\sigma^2} \quad (2)$$

The posterior mean $\mu_1$ can equivalently be expressed as the prior mean $\mu_0$ biased to the likelihood mean $y$ weighted by proportion of prior variance[86]. If prior variance $\tau_0^2$ is high, this results in more adjustment of $\mu_1$ towards $y$ relative to $\mu_0$:

$$\mu_1 = \mu_0 + \frac{\tau_0^2}{\sigma^2 + \tau_0^2}(y - \mu_0) \quad (3)$$

The magnitude of influence of the prior mean on the posterior mean results from the ratio of prior and likelihood variance. Because the single contributions of the variances could hardly be identified in our case and because we assumed both, the likelihood and the prior distributions, to have a constant variance across trials, we replaced the ratio by the parameter α. Thus, we applied the following equation in our models:

$$\alpha = \frac{\tau_0^2}{\sigma^2 + \tau_0^2}; \alpha \in [0,1] \quad (4)$$

$$\mu_1 = \mu_0 + \alpha(y - \mu_0) \quad (5)$$

If $\alpha$ is close to 1, the posterior mean results to be closer to the likelihood mean, whereas a lower value of $\alpha$ reflects more influence of the prior on the posterior mean. The likelihood mean $y$ was defined as the VAS target of the specific trial depending on its intensity $i$ (30, 50, or 70). A linear term $ht$ was subtracted from $y_i$ to account for habituation (or sensitization) over trials $t$.

$$y_t = y_i - (ht) \quad (6)$$

The rating in each trial was modelled as a draw from the normally distributed posterior around $\mu_1$ and a subject-specific posterior standard deviation $\tau_1$.

To explore differences between the conditions, we made some further additions and adjustments to the basic version of the model. First, we needed to account for the unpredictable condition. Based on studies on predictability and pain[27], we theorized that participants average their expectations over the three different intensity levels in the unpredictable condition. We compared two different ways to implement this average expectation in our models.

In the fixed version of the expectation integration models, we defined $\mu_0$ of the unpredictable trials as a draw from the normal distribution around the mean of the 0–100 VAS scale ($N(50, 100)$). Each subject had an individual parameter value for this uncertain prior in the unpredictable trials, but it was kept constant across all unpredictable trials for one individual Table 1.

In the dynamic versions of the model $\mu_0$ was defined as the summed product of outcome probabilities (weights) and prior mean values of the different intensities. The weights ($w$) were defined by dividing the remaining number of occurrences of each intensity by the remaining trial numbers in a run ($\frac{5}{15} = \frac{1}{3}$ for all intensities in trial 1). The prior means $\mu_0$ of each intensity level were then multiplied with their respective weights and added together to form the new unpredictable prior mean of the trial. While all intensities were possible in the first trial, a slight change in outcome probabilities occurs if the same intensity was applied repeatedly. This could lead to a shift in $\mu_0$ for the next trial (e.g., a decrease after experiencing 3 consecutive high stimuli) Table 2.

We first fitted one version of the model, in which we sampled the ratings directly from the prior distribution in the controllable and predictable condition. However, as we were mainly interested in differences between the controllable and the predictable conditions regarding expectation-integration processes, we varied the α parameter across conditions, as it reflects the ratio of prior to likelihood variance. Here, it is important to note, that while it is clear how different prior means across conditions could lead to varying posterior means, differences in prior precision generally affect only the precision of the posteriors, not the posterior means. However, in the context of our pain models, the fixed boundaries of the VAS (ranging from 0 to 100, indicating pain threshold and tolerance) necessitate a different approach. We chose to use truncated normal distributions to

### Table. 1 | Model variables and group-level priors

| Model parameters | Hyperpriors | Description |
|---|---|---|
| $\mu_{0, \text{VAS } 30}$ | $N(30, 100)$ | Prior mean for low intensity |
| $\mu_{0, \text{VAS } 50}$ | $N(50, 100)$ | Prior mean for medium intensity[a] |
| $\mu_{0, \text{VAS } 70}$ | $N(70, 100)$ | Prior mean for high intensity |
| $\log(\sigma_{\mu 0})$ | $N(1.5, 0.5)$ | Standard deviation of prior means |
| $\alpha$ | $N(0.5, 0.3)$ | Scaling parameter |
| $h$ | $N(0, 1)$ | Habituation term |
| $\log(\tau_0)$ | $N(1.5, 0.5)$ | Prior standard deviation |
| $\log(\tau_1)$ | $N(1.5, 0.5)$ | Posterior standard deviation |
| $\mu_1$ | | Posterior mean |
| $w_{\text{VAS } 30}$ | | Probability of low intensity |
| $w_{\text{VAS } 50}$ | | Probability of medium intensity |
| $w_{\text{VAS } 70}$ | | Probability of high intensity |
| Input | | |
| $y_i$ | $\in \{30, 50, 70\}$ | Likelihood mean |
| $t$ | $\in [1,15]$ | Trial number |
| Ratings | $\in [0,100]$ | Pain ratings |

[a]Prior ($\mu_0$) for all intensities in the unpredictable condition in the fixed versions of the models.

### Table 2 | Definition of prior mean in the unpredictable trials

| Model version | $\mu_0$ |
|---|---|
| Fixed | $\mu_{0, \text{VAS } 50}$ |
| Dynamic | $w_{\text{VAS } 30}\mu_{0, \text{VAS } 30} + w_{\text{VAS } 50}\mu_{0, \text{VAS } 50} + w_{\text{VAS } 70}\mu_{0, \text{VAS } 70}$ |

model the pain ratings. In this scenario, the precision of the ratings does impact the resulting means; specifically, less precise ratings tend to produce higher values at lower intensities and lower values at higher intensities because the values are more dispersed across the scale only in one direction and thus via the sampling process are biased in different directions (see Fig. S9).

We started the first analysis approach with splitting up the $\alpha$ parameter. First, we fitted one $\alpha_u$ for the unpredictable condition and a shared $\alpha_{CP}$ for the other two conditions. Still, this model could not differentiate between the controllable and the predictable condition. The more important expansion of the model defined three scaling parameters ($\alpha_u, \alpha_p, \alpha_c$) to differentiate between the controllable, predictable and unpredictable variance ratios. Unfortunately, we were not able to meaningfully recover all the parameters in this model (see Fig. S10).

However, our analysis of the standard deviations of the pain rating data strongly suggested that the observed rating differences between the controllable and predictable condition might actually have resulted from differences in prior variances. So we continued this approach with a leaner model. To test if rather the differences in prior means or prior precision might have produced the interaction between controllable and predictable pain ratings over intensity levels, we went back to our basic model version, with a single $\alpha$ parameter for the unpredictable condition, but applied two different constraints to the flexibility of parameters for the priors in the other two conditions. In the precision-change version of the basic model, $\mu_0$ for all intensity levels was constrained to be equal between the controllable and the predictable condition, but we allowed different standard deviations for the distribution of the pain ratings $\tau_0$. It is important to point out that due to the reductionism of the model, the ratings were in fact sampled directly from the prior. We argue that this is a valid approach, as we assume a likelihood to be constant in all conditions and thus change the shape of the two prior distributions in the controllable and predictable condition in exactly the same way (see Fig. S11).

In the mean-shift version of the basic model, we constrained the standard deviations of ratings $\tau_0$ to be equal, while the prior means $\mu_0$ were allowed to differ between the controllable and predictable condition and for all intensity levels (for an illustration of the difference between precision-change and mean-shift model see Fig. S11). With regard to the unpredictable condition, the fixed and dynamic approaches outlined above were both tested in the mean-shift and the precision-change model. Parameters of both models were recoverable (Figs. S12 and S13).

We compared the models with two null models. The first null model only included the habituation term, while sampling all ratings in all conditions from a normal distribution around the likelihood mean with subject-specific noise. The noise term was sampled on log space with a prior on $N(1.5, 0.5)$.

The second null model estimated prior means of all intensities in the controllable and predictable condition, similar to the mean-shift model version, but did not include the scaling parameter or mean weighting in the unpredictable condition, in which the rating was sampled from the likelihood mean with subject-specific noise.

Model fitting was performed using the MCMC NUTS algorithm implemented in Stan (number of iterations = 4000, number of chains = 4, number of warm up iterations = 1000). Model fitting diagnostics (chain convergence, pairs plots) were examined visually for all models and are summarized in the code repository.

## Parameter recovery of expectation integration models
To be sure that a model can at least theoretically perform as intended, we simulated data of the null models and the different expectation integration models with a range of parameter values in order to perform parameter recovery. Data of 60 hypothetical participants were simulated following the different model types. We decided that the synthetic sample size should match the empirical sample. The different models were fit to the data and parameter values resulting from the fit were correlated with the parameter values entered into the simulation. We visually explored the relationship between the simulated and recovered parameter values and consulted $p$-values and correlation coefficients of the resulting correlations. All parameters could be well recovered for the null models, the mean-shift model, the precision-change model and the model with two $\alpha$ parameters. From the model including three $\alpha$ parameters, $\alpha_p$, was not recoverable (see Fig. S10). We changed the number of iterations to 1000 and the number of warmup iterations to 500 for the parameter recovery analysis.

## Comparison of expectation integration models
Model comparison was done using the robust version of the leave-one-out cross validation (LOO)[87,88]. With LOO, the *expected log pointwise predictive density* (ELPD) for a new data point can be estimated. If the model has a high predictive density across data points, it describes patterns in the data well. The method uses *Pareto smoothed importance sampling* for the estimation of importance weights (the ratio of posterior probability given the data set without a specific data point to its probability given the all data points). Importance weights define how much influence the data point has on the entire posterior probability. An important data point would gain a higher weight in the cross validation procedure. One diagnostic of this method are the Pareto-$\hat{k}$ values, which shape the Pareto distribution that is used to smooth the importance weight distribution. If $\hat{k}$ values are too high, the importance weights cannot be reliably estimated and thus also the results of the cross validation cannot be interpreted. Because this was the case for our models, we used the robust method proposed by Silva and Zanella[87] that employs an alternative mixture estimator to compute the ELPD. We therefore established a robust version for each model. A higher positive value of ELPD indicates a higher probability of the data given the posterior over parameters. We compared the resulting ELPD of the robust models. A difference of ELPD < 4 is considered small. Because the differences between models were larger than 4 and we had more than 100 observations in our data, we could reliably compare the results with the standard error of the difference (see blog post on https://users.aalto.fi/~ave/CV-FAQ.html#se_diff).

To create meaningful parameter values and model output from our winning model, we additionally fitted the non-robust version of the winning model to compare the model predictions to the true rating data of the participant on group- and single-subject level. Exemplary predicted ratings by the models are displayed in Fig. 3) and follow the true ratings.

## Additional analyses of behavioral data
As a readout for different cognitive demand in the different conditions, we analyzed reaction time differences between the controllable, predictable, and unpredictable condition. In controllable trials, reaction time was determined with reference to the time point of the choice. In the predictable and unpredictable trials, the reaction time refers to the time the participants needed to select the correct color from the buttons displayed on the lower part of the screen. Due to time constraint, we ran the reaction time analysis with the lme4[81] package in R, including session number, trial number, and condition as fixed effects and a random intercept for each subject to account for repeated measurements.

## FMRI data acquisition
Structural and functional images were acquired using a PRISMA 3 T MR Scanner (Siemens, Erlangen, Germany) with a 64-channel head coil. A functional MRI sequence of 50 slices (voxel size = 2.0 mm³) was acquired using T2* weighted gradient echo-planar imaging (EPI; TR = 1.5 s, TE = 26 ms, flip angle = 60°, FOV = 224 mm, multiband factor = 2, GRAPPA PAT factor = 2). A structural T1-weighted magnetization-prepared rapid

acquisition gradient echo image was additionally acquired (voxel size 1.0 mm³, 240 slices) before the functional runs. In addition, a B0 field map to correct for magnetic field inhomogeneities was acquired.

### FMRI data analysis

FMRI preprocessing and data analyses were performed using SPM12 (build 7771)[89]. Functional images were slice-timing corrected, realigned, and non-linearly co-registered to the T1-weighted image. For non-linear co-registration the mean EPI and T1-weighted images were segmented. In a next step, nonlinear spatial normalization of the segments of the different tissue classes from the mean EPI to the T1-weighted image was performed. Finally, we computed flow fields to map the T1-weighted images to MNI space (MNI ICBM 152; 2009c Nonlinear Asymmetric[90]). All flow fields for co-registration and normalization were computed with DARTEL[91].

The slice-timing corrected and realigned images were masked with the smoothed (3 mm FWHM smoothing kernel) skull stripped T1-weighted image and used in the first-level analysis. We computed standard GLMs with hemodynamic response function convolved regressors. Onsets were defined at plateau temperature onset of the stimulus (after the rise time of the thermode to reach the temperature). Z-standardized pain ratings were added as parametric modulators. Each boxcar regressor was defined with a duration of 4 s. This resulted in one onset regressor and one modulating parametric regressor for each condition (6 regressors of interest in total). Physiological noise was modeled using the RETROICOR as implemented in the TAPAS toolbox[92] and 24 motion regressors were added to the first-level design matrix. All runs of a participant were concatenated and a session specific intercept was included. After estimating the model, the resulting beta images were warped to MNI space and smoothed with a 4 mm FWHM smoothing kernel.

For the second-level analyses, the normalized and smoothed beta images of the effects of interest were entered into a flexible factorial model in SPM. We separately analyzed pain onsets and parametric modulation of pain by ratings at the group-level with one-way ANOVAs. We computed main effects of pain onset and the condition differences in the onset model and the main effect of intensity in the model including the parametric modulators.

### Additional analyses of fMRI data

To provide the equivalent analysis for fMRI data that we report for behavior, we tested for interactions between intensity levels and condition in brain activity during pain. To implement this analysis, we set up a GLM with separate pain onset regressors for each condition and added the z-scored intensity levels to each pain onset regressors as parametric modulators. We then tested all pairwise interactions between the conditions. No effect was significant at $p < 0.05$ (FWE whole brain correction). Visual inspection of the interaction effects at an uncorrected threshold of $p < 0.001$ revealed multiple activation clusters. As a next step, we investigated, if these activations conform with the direction of the interaction effects present in the pain ratings, by plotting the parameter estimates for each intensity-condition combination (see Supplementary Information, Fig. S14b). As an exploratory post-hoc analysis step, we then evaluated the significance level using an ACC mask generated with the AAL3 atlas[93] and FWE small-volume correction.

### Visualization

Visualization of behavioral data was realized with the ggplot[94] and ggdist[95] packages in R[79]. Neural results were visualized using nilearn[96] in Python 3.9.16.

### Reporting summary

Further information on research design is available in the Nature Portfolio Reporting Summary linked to this article.

## Data availability

The neural and behavioral data generated in this study have been deposited in the GIN (*Modern Research Data Management for Neuroscience*) database and are available at: https://doi.org/10.12751/g-node.fpwjn2.

## Code availability

The scripts to perform the analyses necessary to reproduce the results have been deposited in the following GIN repository: https://doi.org/10.12751/g-node.1qbf71.

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

## Acknowledgements

C.B. is supported by ERC-AdG-883892-PainPersist and by the Deutsche Forschungsgemeinschaft (DFG, German Research Foundation) SFB 289 Project A02 (Project-ID: 422744262–TRR 289).

## Author contributions

Conceptualization, C.B. and M.H.; Methodology C.B. and M.H.; Formal Analysis, C.B. and M.H.; Data Curation M.H.; Project Administration, M.H.; Investigation M.H.; Visualization M.H.; Writing–Original Draft, C.B. and M.H.; Funding Acquisition C.B.; Resources C.B.; Supervision C.B.

## Funding

## Competing interests

The authors declare no competing interests.
