## [Transparent Peer Review file · Nature Communications]

Controllability Changes Pain Perception by Increasing the Precision of Expectations

Corresponding Author: Ms Marie Habermann

Version 0:

Reviewer comments:

Reviewer #1

(Remarks to the Author)

This paper reports on two independent experiments with two different samples that are aimed to test the effects of the possibility to exert control over painful stimuli on the experience of those, regardless of the possibility to predict these stimuli. The study is important as the literature reveals a large heterogeneity of results, (with increases, no change, and increases in pain intensity). In that sense, the study is welcome as it addresses a critical literature gap characterized by substantial result heterogeneity—encompassing variable pain intensity responses potentially attributable to predictability's confounding influence.

This study aimed to disentangle the effects of predictability and controllability at both the behavioral and neural levels. The results show that the modulatory function of control was parsimoniously explained by an increase in expectation precision, and absent in a condition with equal predictability but where control was not possible. This suggests that control induced an increase in outcome certainty, rather than predictability, which leads to pain modulatory effects.

The studies are elegantly set up, the methods are the state of the art and clearly outlined, the analyses appropriate, and the results are clear and well discussed.

Despite these strengths, I do have several comments that the authors may consider.

1. Dimensionality of Control: Researchers have identified several distinct dimensions of control: control over onset (predictability), duration, frequency, location, spatial, sensory characteristics, contextual, intensity, and offset (escape) of the aversive stimuli, to name the most important. In their controllable condition the participants chose a stimulus intensity at the beginning of each trial. There was a predetermined number of 15 stimuli of different intensities (low, medium, high) that had to be delivered to the participants, and they only had control over the order by which they were delivered. Do the authors think their results can be generalized to the other dimensions of control? I suggest the devote a couple sentences about external validity in the discussion part of the paper.

2. Experimental Design Limitations: In the predictable and the uncontrollable conditions, the participants were either informed about the upcoming stimulus intensity or not at all. To control for potential confounding effects of stimulus order on the primary outcome measure of pain intensity ratings, a "yoked" experimental design represents an optimal methodological approach. In such a design, one group's choices (order of the stimuli) are applied to the other two groups. While this methodological control appears to have been implemented in the second study, its absence in the first study compromises the internal validity and interpretability of the experimental findings. Please clarify.

3. Behavioral Observations: The authors found that individuals tended to select highly painful stimuli sooner rather than later, and that the reason for this was to alleviate dread and overall discomfort. Regarding the latter, did the authors measure this, or is it an assumption? Alternatively, could this finding also be the results of exploratory behavior of the participants?

4. Neuroimaging Findings: Neural activity during predictable and controllable pain. Acknowledging that the authors are internationally renowned experts in neuro-imaging, I am still wrestling with the meaning of their fMRI findings. Are they just providing converging evidence supporting then validity of the behavioral data, or do they reveal a unique finding that would not have been discovered without these (expensive) imaging data? It would be interesting for the reader if the authors could

devote some sentences about this in the discussion section of the manuscript.

5. Minor conceptual Critique: The authors' assertion that control over aversive stimuli can lead to an active coping style presents a potential tautological construct. The proposed causal relationship appears conceptually problematic, as "active coping style" is frequently operationalized, in part, by the capacity to exert control over aversive stimuli. Consequently, the proposed mechanism risks circular reasoning, wherein the explanatory variable is intrinsically embedded within the definition of the outcome variable. Moreover, the broader construct of "coping" itself remains theoretically ambiguous, characterized by significant conceptual heterogeneity and definitional imprecision.

I enjoyed reading this manuscript

(Remarks on code availability)

Reviewer #2

(Remarks to the Author)

In the current study, Habermann and Büchel address an interesting question – how controllability modulates experience of pain. The authors use a novel experiment in which participants rate painful stimuli in three block types: controllable (C), predictable (P), and unpredictable (U). In controllable trial types, the participants can choose the next stimulus intensity that they get from a set of three options. In the predictable condition, participants do a pain-irrelevant task (color matching), but they observe the next stimulus intensity that they are about to receive in the next trial. Finally, in the unpredictable trials, participants only do the color matching task, and do not know the incoming stimulus intensity type of the next trial. By looking at the pain ratings in different stimulus intensity levels and in different tasks, the authors show that there is an interaction between predictability and pain intensity, where the low stimulus intensity, the pain rating is lower for controllable and predictable outcomes compared with the unpredictable one, while in the high pain intensity levels, the pain rating is higher for controllable and predictable conditions compared with unpredictable ones. Using computational Bayesian modeling and brain imaging, the authors provide evidence for an increase in the precision of expectations in controllable condition, compared with predictable and unpredictable conditions.

The study makes an interesting and novel claim, that controllability effects on pain are a result of increased precision in prior beliefs about pain intensity, rather than a shift in prior beliefs about the level of pain intensity. The concept of comparing controllable and uncontrollable stimuli while matching on predictability is an innovative one, and the topic is a perennially important one in the field. In addition, the fMRI results show tantalizing reductions in the controllable vs predictable conditions in a variety of important areas.

While interesting, there are several potential confounds and the conclusions may depend strongly on modeling assumptions. They also seem to diverge from both the models tested and conclusions reached by previous related work from the same lab, leading to some questions, which are elaborated below. We don't seem to be able to tell from this experiment whether control is analgesic, or assess the causal effects of controllability on pain. The behavioral analyses address interactions with stimulus intensity only, whereas the brain analyses examine contrasts across all stimulus levels. There are some potential confounds with cognitive demand that could influence the brain findings (and secondarily pain levels). On balance, it seems important to address these issues before we can be confident in the study's conclusions.

Potential confounds

One concern is a potential confound with cognitive demand, which some earlier experiments (e.g., Salomons et al.) also suffered from. The color-matching task is likely easier and produces faster RTs than the controllability task, and this cognitive demand effect could bleed over into the subsequent stimulus processing period, explaining reductions with controllability in SMA, ACC, and other areas. It would be helpful to analyze and present RTs for the various conditions. It would also be helpful to assess choice-related (pre-stimulus) activity differences for controllable vs. predictable stimuli. Differential activity in the same regions could signal a confound due to vascular nonlinearities (it is harder to increase BOLD signal in an area that has been recently activated). The amount of jitter here is likely too low to prevent potential confounding.

For the estimation of control effects to be valid, the intensity of stimuli must be successfully matched, and sequence effects matched as well, particularly when analyzing data as a function of stimulus intensity (as with the behavioral analyses). It would be helpful to carefully assess the degree to which the stimulus and transition probabilities were matched across controllable and predictable conditions. It may also be helpful to control for stimulus intensity when assessing controllability effects in the brain, as this would help control for residual mis-matching. Sequence differences present a potential confound, as they could differentially affect high and low intensity trials across controllable and predictable conditions. The fact that the method for matching was altered mid-way through the experiment underscores the need for careful assessment here.

Statistics

I have a few suggestions on the statistics. The reported posterior credibility intervals (HDPs) are somewhat difficult to interpret, because it's hard to be sure whether some of the effects are likely due to chance and how large the effects are in terms of standard effect sizes. The posterior intervals are being treated as equivalent to frequentist confidence intervals from the perspective of inference (conclusions and ***s on plots, etc.) But one can make the Bayesian credible interval more or

less tight depending on the choice of priors. Can you show that with the selected flat priors the credible interval provides the expected coverage, so that these intervals can indeed be interpreted like standard frequentist CIs?

Pp 4-5. It seems that statistics are reported on the behavioral-only sample, and then selectively for the fMRI sample, without reporting full statistics for each sample in a way that clearly demarcates them.

Finally (on the linear models), it's not clear what the random effects structure is for these models and the degrees of freedom. Which effects were modeled as random? I don't believe random slopes for experimental conditions are mentioned, but they would be appropriate here. It's difficult to interpret the results without this information.

Analyses included and omitted

I expected to see an analysis and report of the main effects of controllability compared with unpredictable/uncontrollable and predictable compared with unpredictable/uncontrollable. It looks as though there may be no effect of control overall, apart from providing an anchor for expectations. We don't seem to be able to tell from this experiment whether control is analgesic, or assess the causal effects of controllability on pain. This ambiguity is puzzling, as the brain contrasts analyze exactly these kinds of main effects across all trials. So the behavioral data show bias towards predicted values, with controllable and predictable conditions both offering strong value predictions, whereas the brain effects show a different effect, the reduction (or increase) in pain-linked brain activity due to predictability and controllability overall. It seems important to analyze both these overall (main) effects and interactions (bias towards predicted levels) in both behavior and brain, and understand how they are linked.

The main behavioral finding is that the standard deviations of ratings were lower in the controllable condition than in the predictable condition in the fMRI sample. The authors attribute this to an effect of control, as the stimulus intensities are nominally the same. But inferential statistics are not provided.

The analyses assume the mean shift or precision change is consistent (the same mechanism) in both tasks. What if one task has mean shift mechanism and the other has precision modulation? Also, what if both mechanisms are involved? Is it possible to expand the model comparisons to test these possibilities?

The analyses assume that shifts in mean and variance (precision) are independent, and that a mean shift will entail no change in the variance of its associated distribution. However, this is a strong assumption that may not be warranted. The means and variances of measured phenomena are often linked. If so, the differences between the controllable and predictable conditions could be due to differences in mean levels for controllable and uncontrollable rather than qualitative differences in precision. The standard deviation, which is used to estimate precision, varies with the mean and cannot be assumed to be constant across the range of the rating scale. Can you provide evidence that the mean and precision are independent here, and potentially explore models in which they are not strongly decoupled?

What features of the data are driving the differences in model fit between the mean shift and precision shift models? It is not clear how different the predictions of the two models are in this case. This is interesting as the results appear to conflict with the type of effect found in similar recent work by Strube et al. from the same lab. It is also important to understand because of the approximations used here (both use of Gaussian distributions and approximate matching of the stimulus intensity and sequence of intensities) and the potential for shifts in mean ratings to be linked to shifts in the variance of ratings in ways the models do not account for.

Theoretical implications

The findings are cast in terms of validating Bayesian theories of pain, but the authors may want to word their intro and discussion carefully, because it is not clear that the Bayesian model makes strong predictions here that are not made by other theories, and thus the paper doesn't really "prove" that the Bayesian model is the "right" one. Existing theories in psychology suggest that expectations influence perception, causing biases in perception towards predicted values (or other sources of information), which produces the kinds of interaction effects found here. Even simply adding more noise in judgments in the unpredictable condition (and in the predictable relative to the controllable condition) would produce the same qualitative pattern of interactions. The paper also doesn't attempt to compare Bayesian with non-Bayesian models. The Bayesian framework is very useful and sensible, but I would feel more comfortable if these results were not presented as a strong validation of Bayesian theories.

This paper seems to be inconsistent at a theoretical level with Strube and Buchel's work from the same lab. There, they test an overlapping construct, agency, and conclude that it results from a shift in priors (lower pain expectations). They test a likelihood precision model, positing that even though the stimulus intensities are the same for agency and no-agency conditions, the likelihood precision may differ. Here, however, it is claimed that the likelihoods must in principle be the same across control and no-control conditions because the stimulus intensities are the same. This seems like a strong and unwarranted assumption given the prior work. It seems possible that the experiments do not have the necessary conditions to disentangle the various possibilities. For example, a higher prior precision with control could in fact be a lower posterior precision, and this may explain the same effects. If such ambiguities are present, it would be helpful to explain them transparently.

The alpha parameter seems to be a non-Bayesian replacement for the precision of the stimulus, and it is unusual in that it seems to be outside the Bayesian framework the paper purports to test. Why not actually parameterize the precision of the stimulus (or relative precision of prior and stimulus)?

Other clarifications

The authors suggest that control increases outcome certainty, more than mere predictability. The distinction between certainty and predictability, while mentioned, could benefit from additional clarification for non-specialist readers. Further, in line with the first point that was raised, I am not convinced that controllability increases outcome certainty in general – though it does in this paradigm -- especially if the outcomes are probabilistic and influenced by multiple factors. It may be helpful to discuss that controllability may be differentially associated with predictability across different contexts.

The behavioral findings on predictability are described as though predictability causes a complex effect of decreases in low-intensity and increases in high-intensity stimuli (see Results and Discussion). While technically an interaction effect, the language obscures the likely story by casting predictability as a causal construct. A more parsimonious explanation is that ratings are biased towards their predicted values, so if a person expects a low-intensity stimulus with high confidence (high precision), they report lower intensity, and if a person expects a higher-intensity stimulus, they report higher intensity. I think the authors likely agree with this explanation, as it fits the Bayesian framework, but the writing could tell this story more clearly.

Other points

PAG and aversive prediction errors: Is it unpredictability and/or lack of control that increases PAG activity, or is it worse-than-expected stimuli (aversive prediction errors) specifically?

In Fig 5 C/D it does not look like any regions survived multiple comparisons, as there are no areas in black outlines as in 5A/B?

Please describe the anchors used for VAS ratings (e.g., of 0 and 100), and summarize with statistics the stimulus intensity for threshold and tolerance, and how these map onto VAS ratings.

(Remarks on code availability)

Reviewer #3

(Remarks to the Author)

(Remarks on code availability)

Version 1:

Reviewer comments:

Reviewer #1

(Remarks to the Author)

The authors have done an outstanding job in addressing the reviewers' comments. They not only provided additional information but also conducted further analyses and included new figures, all of which are highly convincing.

However, a more fundamental scientific question remains in the background—though it does not detract from my overall positive assessment of the manuscript. This pertains to the added value of the behavioral versus the imaging results. While it is reassuring that the fMRI findings align with the hypothesis (which is also supported by the behavioral data) that control effects are associated with increased expectation precision, the authors' response—"We think that the imaging data is crucial to validate the model by bringing forward the neurophysiological processes possibly underlying the proposed mechanisms"—does not entirely resolve my concerns. Specifically, had the imaging data not fully corroborated the behavioral results, how problematic would this discrepancy have been?

This issue brings to mind Gregory Miller's (2010) seminal paper, which eloquently discusses the challenge of defining what constitutes an adequate explanation in the relationship between psychological and biological phenomena. The paper cautions against assuming direct causal links between neural and psychological events, as they operate at different levels of analysis. If they agree, the authors may wish to add a sentence or two in the discussion section of the manuscript.

(Remarks on code availability)

N.A.

Reviewer #4

(Remarks to the Author)

I am reviewing the revised manuscript on behalf of Reviewer #2, who provided a comprehensive review of the original manuscript but is currently unavailable for further comments. I concur with Reviewer #2 regarding the value of the study,

particularly the observation that “the study makes an interesting and novel claim” and that comparing controllable versus uncontrollable conditions while matching on predictability is “an innovative” approach. The topic is indeed “a perennially important one,” as noted.

Reviewer #2’s comments were thorough, and some of them touched on fundamental issues. However, I find that the authors have responded to all points both elegantly and comprehensively. They conducted multiple additional analyses, which have been included in the supplementary information. In my view, the reviewer’s feedback and the authors’ revisions have substantially improved the manuscript. I therefore recommend that the paper be accepted in its current form.

(Remarks on code availability)

The images or other third party material in this Peer Review File are included in the article’s Creative Commons license, unless indicated otherwise in a credit line to the material. If material is not included in the article’s Creative Commons license and your intended use is not permitted by statutory regulation or exceeds the permitted use, you will need to obtain permission directly from the copyright holder.

Reviewer #1

(Remarks to the Author):

This paper reports on two independent experiments with two different samples that are aimed to test
the effects of the possibility to exert control over painful stimuli on the experience of those, regardless
of the possibility to predict these stimuli. The study is important as the literature reveals a large
heterogeneity of results, (with increases, no change, and increases in pain intensity). In that sense, the
study is welcome as it addresses a critical literature gap characterized by substantial result
heterogeneity—encompassing variable pain intensity responses potentially attributable to
predictability's confounding influence.

This study aimed to disentangle the effects of predictability and controllability at both the behavioral
and neural levels. The results show that the modulatory function of control was parsimoniously
explained by an increase in expectation precision, and absent in a condition with equal predictability
but where control was not possible. This suggest that control induced an increase in outcome certainty,
rather than predictability, which leads to pain modulatory effects.

The studies are elegantly set up, the methods are the state of the art and clearly outlined, the analyses
appropriate, and the results are clear and well discussed.

Despite these strengths, I do have several comments that the authors may consider.

**Dimensionality of Control:** Researchers have identified several distinct dimensions of control: control
over onset (predictability), duration, frequency, location, spatial, sensory characteristics, contextual,
intensity, and offset (escape) of the aversive stimuli, to name the most important. In their controllable
condition the participants chose a stimulus intensity at the beginning of each trial. There was a
predetermined number of 15 stimuli of different intensities (low, medium, high) that had to be delivered
to the participants, and they only had control over the order by which they were delivered. Do the authors
think their results can be generalized to the other dimensions of control? I suggest the devote a couple
sentences about external validity in the discussion part of the paper.

We agree on the broad dimensionality of control. While it is true that principally the sequence was
controllable, this enabled participants to flexibly choose among the intensities in most of all trials.

Indeed, other types of control might involve additional mechanisms, particularly when control is
exercised over the initiation of inherently aversive stimuli (such as the application of a painful stimulus)
in contrast to positive events (like the initiation of pain-relieving treatment). Additionally, in situations
where self-motion is directly related to the aspect under control (e.g., removing a splinter), processes
related to sensory attenuation processes may also come into play, leading to a general reduction in
sensory input during self-touch, possibly depending on the temporal gap between action and outcome.

We believe that the increase in expectation precision could be a significant factor for *all types* of
control, with respect to the feature of the stimulus over which control is exerted. For example, when
the onset is controllable, the expectation of the stimulus start becomes more precise; and when its
intensity is controllable, the expectation regarding its intensity becomes more precise.

Therefore, our results are potentially generalizable to other dimensions of control, even if the increase
in expectation precision (or subjective predictability) can yield different outcomes regarding perceived
pain, depending on the specific control dimension being examined, as illustrated by the interaction
effect observed in our study. For example, control might amplify the perceived effectiveness of a

treatment or lead to a reduction in pain when it aligns with positive expectations, whereas it may result
in higher pain ratings when control is exercised over the onset of a highly painful stimulus. These results
would be main effects, if only one intensity level is investigated. Our study highlights the importance
of considering both predictability and outcome valence or if possible, test effects across a range of
intensities.

We have included additional sentences addressing external validity in the Discussion section of the
manuscript.

**Experimental Design Limitations:** In the predictable and the uncontrollable conditions, the
participants were either informed about the upcoming stimulus intensity or not at all. To control for
potential confounding effects of stimulus order on the primary outcome measure of pain intensity
ratings, a “yoked” experimental design represents an optimal methodological approach. In such a design,
one group's choices (order of the stimuli) are applied to the other two groups. While this methodological
control appears to have been implemented in the second study, its absence in the first study compromises
the internal validity and interpretability of the experimental findings. Please clarify.

We agree that a yoked between-participants design is an approach that allows for precise matching of
stimulus sequences across participants. However, to minimize between-subject error variance we
preferred a within-participants design. Unfortunately, a fully yoked within-participant design does not
allow for full counterbalancing of conditions (controllable runs would always need to be completed
before their predictable and unpredictable counterparts). Additionally, using the same sequence for
both the predictable and unpredictable conditions could reduce the effects of unpredictability.

While we could create similar stimulus sequences in the second study, we did not know the average
choice behavior in the controllable condition when we started with study 1 (behavioral sample).
Consequently, we opted for a randomly uniform distribution of stimulus order in study 1.

During the quality control stage of data analysis of the first study, we observed that participants tended
to select high pain at the beginning of a run and reserved lower pain stimuli for the end. Therefore, we
adapted our method for sequence matching in study 2 (fMRI sample).

To demonstrate the improvement of our matching procedure, we provide a time-resolved analysis of
frequency differences of high, medium, and low intensity stimuli between the conditions. Our findings
indicate a significant improvement in the matching procedure for the second study (Figure A). We also
included this figure in the supplemental information of the paper.

 **Figure A** | The figure shows the within-participants differences between stimulus intensity frequencies in both
 runs grouped in three bins of 5 trials, separate for each intensity level. We grouped the data in three time bins
 (trials 1:5, 6:10, 11:15) and then subtracted, for each time bin the frequency of predictable from controllable
 stimuli of each intensity level. A positive value indicates a higher occurrence frequency of that intensity level in
 controllable trials, and a negative value indicates that the intensity level occurred more frequently in predictable
 trials. As expected, the high intensity was more frequent in the controllable condition in the first time bin, and
 that the low intensity was more frequent in last third in the behavioral sample. In the fMRI sample, where the
 frequencies were better matched, we see that in all trial bins the differences in frequency are well distributed
 across participants and unlikely to have introduced a systematic confounding effect and indicating improvement
 in matching procedure

**Behavioral Observations:** The authors found that individuals tended to select highly painful stimuli
sooner rather than later, and that the reason for this was to alleviate dread and overall discomfort.
Regarding the latter, did the authors measure this, or is it an assumption? Alternatively, could this
finding also be the results of exploratory behavior of the participants?

While we did not measure dread or discomfort with ratings or questionnaires, our conclusions are
theoretically motivated by the existing literature that has tested models exploring the effect of
negative anticipation that we have effectively replicated in our study.

Because we provide participants with control over the entire sequence of stimuli, we believe they will
converge to the behavior that is most "comfortable" for them. Selecting high pain at the beginning
seems to reduce the time spent "worrying" about how painful it might be. After experiencing the first
high pain, participants can quickly reach the end of that experience, effectively minimizing the time of
negative anticipation, which we refer to as "dread". While "dread" may seem like an extreme term for
this phenomenon, it is derived from several studies that have demonstrated a similar effect: waiting
for future pain becomes increasingly aversive, and choosing the option of pain relief is often delayed,
as shown in previous studies (Story et al. 2013, 2015).

In addition, this reviewer is correct in assuming that exploratory behavior of participants also plays a
role in the individual choice patterns. This can be seen as a small "spike" in the histograms of high, low,
and medium pain during the first three trials (see Supplemental Figures S1a and S2a). We believe that
apart from the effects induced by negative anticipation and dread alleviation, i.e., choosing high pain
first to confront the worst, participants probably also explored the other pain intensities in the first
trials to make informed choices for the remainder of the run. We added this to the description of the
Supplemental Figures showing the intensity sequences in the different conditions and samples (S1, S2).

While exploration was present, we think it cannot explain the stronger pattern of choosing high pain
at the beginning of a run, especially since participants completed two runs of the controllable
condition, which reduced their need for exploration in the second run. We believe this pattern mainly
results from selecting the most comfortable sequence which seems to be influenced by dread.

**Neuroimaging Findings:** Neural activity during predictable and controllable pain. Acknowledging that
the authors are internationally renowned experts in neuro-imaging, I am still wrestling with the meaning
of their fMRI findings. Are they just providing converging evidence supporting then validity of the
behavioral data, or do they reveal a unique finding that would not have been discovered without these
(expensive) imaging data?

It would be interesting for the reader if the authors could devote some sentences about this in the
discussion section of the manuscript.

The fMRI findings support the hypothesis, which is also supported by the behavioral data, that the
effects of control are associated with an increase in expectation precision. The activation changes in
salience and attention processing areas when transitioning from unpredictable to predictable and
controllable pain, namely that higher uncertainty in the unpredictable condition induces higher
activation in salience and attention processing areas, are associated to the change in expectation
precision, that also explains the behavioral effects. Importantly, we also observed a significant
difference between the controllable and predictable conditions. This observation underscores our
hypothesis that the internal predictability increases when individuals control the pain intensity, and
that brain activity represents this effect. Thus, the fMRI findings are supportive and consistent with
the behavioral results.

The notion of increased expectation precision is theoretically grounded in an expectation-sensation
integration model of perception (specifically, the Bayesian pain model) and we think that the imaging
data is crucial to validate the model by bringing forward the neurophysiological processes possibly
underlying the proposed mechanisms. In addition, our fMRI data support earlier accounts of Bayesian
integration processes in the brainstem. Increases in BOLD signal in the PAG have been associated with
lower expectation precision (Grahl et al., 2018), an effect that is consistent with our finding that PAG
activity is relatively higher in both the unpredictable and predictable conditions compared to the
controllable condition. Thus, we assert that our findings would be less impactful without the support
of the fMRI data.

Moreover, another aspect addressed in the paper also gains relevance by reporting the imaging data.
While we consider the most interesting effect to be the dissociation of controllability and
predictability, the behavioral and neural data clearly indicate that controllability and predictability
produce many significantly overlapping effects when compared to the unpredictable and
uncontrollable condition. Therefore, the fMRI data is essential to emphasize the need for appropriate
control conditions when investigating control-induced pain modulation. It is crucial that the control
condition renders the stimulus feature under control similarly predictable as in the experimental
condition, to correctly attribute the observed effects to control rather than expectation precision or
predictability.

We added this reasoning to the Discussion section to stress the complementary nature of the neural
and behavioral findings.

**Minor conceptual Critique:** The authors' assertion that control over aversive stimuli can lead to an
active coping style presents a potential tautological construct. The proposed causal relationship appears
conceptually problematic, as "active coping style" is frequently operationalized, in part, by the capacity
to exert control over aversive stimuli. Consequently, the proposed mechanism risks circular reasoning,
wherein the explanatory variable is intrinsically embedded within the definition of the outcome variable.
Moreover, the broader construct of "coping" itself remains theoretically ambiguous, characterized by
significant conceptual heterogeneity and definitional imprecision.

Thank you for bringing this aspect to our attention. We acknowledge that the point we intended to
convey in this sentence was not very clear and could lead to a misunderstanding. What we meant to
express is that control, which may be granted by the environment, is only meaningful if it can be
detected and acted upon, particularly in contrast to a perceived *lack of control* that may exist, even
when strategies are available to improve the situation (e.g., in situations of learned helplessness). We
changed the sentence in the introduction accordingly.

I enjoyed reading this manuscript

Thank you very much, we very much enjoyed writing it 😊

**Reviewer #2**

(Remarks to the Author):

In the current study, Habermann and Büchel address an interesting question – how controllability
modulates experience of pain. The authors use a novel experiment in which participants rate painful
stimuli in three block types: controllable (C), predictable (P), and unpredictable (U). In controllable
trial types, the participants can choose the next stimulus intensity that they get from a set of three
options. In the predictable condition, participants do a pain-irrelevant task (color matching), but they
observe the next stimulus intensity that they are about to receive in the next trial. Finally, in the
unpredictable trials, participants only do the color matching task, and do not know the incoming
stimulus intensity type of the next trial. By looking at the pain ratings in different stimulus intensity
levels and in different tasks, the authors show that there is an interaction between predictability and
pain intensity, where the low stimulus intensity, the pain rating is lower for controllable and
predictable outcomes compared with the unpredictable one, while in the high pain intensity levels, the
pain rating is higher for controllable and predictable conditions compared with unpredictable ones.
Using computational Bayesian modeling and brain imaging, the authors provide evidence for an
increase in the precision of expectations in controllable condition, compared with predictable and
unpredictable conditions.

The study makes an interesting and novel claim, that controllability effects on pain are a result of
increased precision in prior beliefs about pain intensity, rather than a shift in prior beliefs about the
level of pain intensity. The concept of comparing controllable and uncontrollable stimuli while
matching on predicability is an innovative one, and the topic is a perennially important one in the
field. In addition, the fMRI results show tantalizing reductions in the controllable vs predictable
conditions in a variety of important areas.

While interesting, there are several potential confounds and the conclusions may depend strongly on
modeling assumptions. They also seem to diverge from both the models tested and conclusions
reached by previous related work from the same lab, leading to some questions, which are elaborated
below. We don't seem to be able to tell from this experiment whether control is analgesic, or assess
the causal effects of controllability on pain. The behavioral analyses address interactions with stimulus
intensity only, whereas the brain analyses examine contrasts across all stimulus levels. There are some
potential confounds with cognitive demand that could influence the brain findings (and secondarily
pain levels). On balance, it seems important to address these issues before we can be confident in the
study's conclusions.

**Potential confounds**

One concern is a potential confound with cognitive demand, which some earlier experiments (e.g.,
Salomons et al.) also suffered from. The color-matching task is likely easier and produces faster RTs
than the controllability task, and this cognitive demand effect could bleed over into the subsequent
stimulus processing period, explaining reductions with controllability in SMA, ACC, and other areas.
It would be helpful to analyze and present RTs for the various conditions.

It would also be helpful to assess choice-related (pre-stimulus) activity differences for controllable vs.
predictable stimuli. Differential activity in the same regions could signal a confound due to vascular
nonlinearities (it is harder to increase BOLD signal in an area that has been recently activated). The
amount of jitter here is likely too low to prevent potential confounding.

This is a conceptually interesting aspect that potentially affects many fMRI studies that use sequential
paradigms (e.g. dopamine related effects in expectation-outcome studies in the reward domain). The
reviewers refer to adaptation effects in fMRI activity, i.e. the phenomenon that a region that has
recently be activated shows reduced responding to a second task if it is sensitive to shared features
between tasks (Barron et al., 2016). Such adaptation, or repetition suppression, effects have mainly
been shown for the processing of visual stimuli, where they occur at rather short time scale. For
example, showing the same visual stimulus twice with an ISI of 1s, the effect of repetition suppression
leads to an activity reduction of about 10% in lower and higher order visual processing areas (Fritsche
et al., 2020). However, other papers argue that higher order brain regions can show adaption effects
up to ~6s (Klein-Flügge et al., 2013). The net effects of repetition suppression, resulting in adaptation,
and/or even repetition enhancement on BOLD signals are currently a matter of debate, as it is
complicated to dissociate the effect of neural activation and hemodynamic effects (Barron et al., 2016;
Segaert et al., 2013). For example, it has been reported that pre-stimulus activity in bilateral insula
before pain can predict a higher behavioral response to the painful stimulus concurrently with an
activity increase that correlated with the intensity (Boly et al., 2007).

Our tasks (deliberately) differ in terms of cognitive demand, resulting in longer reaction times for
choices made in the controllable condition (mean±SD = 1.76s±0.84s) compared to the reaction in the
color-matching task in the other two conditions (predictable: mean±SD = 1.46s±0.87s; unpredictable:
mean±SD = 1.36s±0.84s) in both studies (fMRI sample: controllable: mean±SD = 1.72s±0.66s;
predictable: mean±SD = 1.09s±0.47s; unpredictable: mean±SD = 1.02s±0.45s) (see Figure B). This
underlines that participants made deliberate choices in the controllable condition, which inherently
take longer.

Figure B | Reaction times were significantly longer in the controllable condition than in the predictable condition and unpredictable condition. sample 1 = behavioral sample; sample 2 = fMRI sample. Analyses were run using lme4 in R, Models included session number, trial number and condition as fixed effects and a random intercept for each subject to account for repeated measurements.

The temporal gap between the choice period and the trigger impulse to start the thermode was in total 9-12s (4s choice, 3s information and 2-5s anticipation) plus an additional delay of ~1s (not shown in Figure 1) until the thermode reached the desired temperature (0.89s for VAS 30, 1.02s for VAS 50 and 1.16s for VAS 70) leading to an effective stimulus onset asynchrony of 10-13s (maybe Figure 1 was a bit misleading and created the impression that this period only lasted for 2-5s).

Nevertheless, we performed the suggested (time-resolved) fMRI analysis to investigate whether the decision task leads to BOLD effect differences at pain onset. We therefore modeled the decision phase with a finite-impulse-response (FIR) model, starting at task onset and covering a total of 16.5s (i.e. 11 bins of 1 TR each). At the beginning of this period, activity in the SMA and PAG was increased during controllable compared to predictable trials (Figure Cb). The peak at bin 4 (4.5-6s) indicates a correct specification for the onset of the task in both regions. We then compared activity at the relevant time bins as modeled in our main analyses for pain onset (i.e. bins 8 & 9 covering 10.5-13.5s). This analysis revealed no significant difference between the conditions in both the SMA and the PAG (see Figure Cc). This further suggests that it is unlikely that these effects are driven by pre-stimulus activity.

We think that this conceptual point is interesting and thus have added it to the Discussion and included the figures in the supplemental information as Figure S6 (RT) and S9 (FIR analysis).

a

b

c

**Figure C | Parameter estimates derived from an FIR model of the decision task.** (a) Shows the time period
covered by the FIR model: 11 TR à 1.5s, resulting in 16.5s in total. Time courses are shown for the peak voxels in
(b) SMA and (c) PAG, where we find a significant reduction in the controllable condition compared to the
predictable condition during pain (see Figure 5b in revised manuscript).

For the estimation of control effects to be valid, the intensity of stimuli must be successfully matched,
and sequence effects matched as well, particularly when analyzing data as a function of stimulus
intensity (as with the behavioral analyses). It would be helpful to carefully assess the degree to which
the stimulus and transition probabilities were matched across controllable and predictable conditions.

It may also be helpful to control for stimulus intensity when assessing controllability effects in the
brain, as this would help control for residual mis-matching. Sequence differences present a potential
confound, as they could differentially affect high and low intensity trials across controllable and
predictable conditions. The fact that the method for matching was altered mid-way through the
experiment underscores the need for careful assessment here.

We thank the reviewer for the comment and agree that matching the stimulus sequences are relevant
in our design. We decided against a complete yoked design, because we wanted to allow full
counterbalancing of conditions to control for habituation effects. In addition, we were naïve about
patterns of choice behavior in the controllable condition before conducting the first study with the
behavioral sample. Indeed, as can be seen in the supplementary figures, the intensity sequences were
not perfectly matched in study 1, due to the shared pattern in choice behavior between participants.
Consequently, we adapted our procedure to enable a better matching between conditions.

We now show a detailed overview of stimulus frequency differences (within subject) in three time bins
over a run as a measure of improvement of the matching strategy. First, to get a within-subject readout
of intensity frequency differences between the controllable and predictable condition, we grouped the
data in three time bins (trials 1:5, 6:10, 11:15). We then subtracted, for each time bin the frequency of
predictable from controllable stimuli of each intensity level. Consequently, a positive value indicates a
higher occurrence frequency of that intensity level in controllable trials, and a negative value indicates
that the intensity level occurred more frequently in predictable trials. As expected, the plots for the
behavioral sample show, that in the first time bin, the high intensity was more frequently chosen in
the controllable condition, and that the low intensity was more frequently chosen in the last third. In
the fMRI sample, where the frequencies were actually matched, the differences in frequency are well
distributed across participants and unlikely to have introduced a systematic confounding effect (see
Figure D). We also included this figure in the supplemental information of the paper.

**Figure D** | The figure shows the within-participants differences between stimulus intensity frequencies in both
 runs grouped in three bins of 5 trials, separate for each intensity level. While it is obvious that the high intensity
 was more frequent in the controllable condition in the first third of a run in sample 1, while the lowest intensity
 was more frequent in the last third, this pattern is absent in sample 2, where the deltas in all three bins vary in
 both directions. *Note.* The difference is computed across both runs, so each intensity x condition combination
 occurred 10 times.

**Statistics**

I have a few suggestions on the statistics. The reported posterior credibility intervals (HDPIs) are
somewhat difficult to interpret, because it's hard to be sure whether some of the effects are likely due to
chance and how large the effects are in terms of standard effect sizes.

The posterior intervals are being treated as equivalent to frequentist confidence intervals from the
perspective of inference (conclusions and ***s on plots, etc.).

But one can make the Bayesian credible interval more or less tight depending on the choice of priors.
Can you show that with the selected flat priors the credible interval provides the expected coverage, so
that these intervals can indeed be interpreted like standard frequentist CIs?

We appreciate this comment, because it led us to clarify a few points and to improve the figures for
an easier access to the underlying statistics. We apologize for the confusion created using stars in the
figures 2 and S4. The HPDIs can in fact not be interpreted as standard CIs and we did not intend to
treat HPDIs as equivalent to frequentist confidence interval from perspective of inference.

The *** measure on the plots were chosen to improve readability of the results, but the similarity to
the indication of significant p -values usually used by frequentist analyses might have caused unwanted
reference to CI and p -values. Apart from the question of equivalence regarding CI and HPDI, the
reviewers raise in their comment the question of effect sizes in Bayesian analysis and the choice of
priors.

Frequentist analyses rely on null hypothesis testing and derive p -values and confidence intervals (CI)
to quantify how likely results were produced by chance and to make inferences about an unknown
population parameter value. A small p -value indicates that the probability for the estimated parameter
to have a certain value is low, under the assumption of a null distribution. But this is not a probability
of a parameter value and unrelated to the probability of the hypothesis of interest (i.e. H_1).

Regarding CIs, it is true that a narrow CI indicates that a sample parameter is likely to be close to a
hypothesized true population parameter and therefore provides some level of inferential information
(Field et al., 2012, p. 48). However, CIs do *not* indicate an uncertainty in the true parameter value by
probability, i.e. that the probability that the true parameter values lie in the interval is 95% (McElreath,
p. 58). Instead the standard CI is the interval in which, if doing many repetitions of the same
experiments and constructing the CI around the sample mean, the true unknown population mean
would fall in 95% of all repetitions, i.e. in 95 of 100 experiments the CI around the mean contains the
"true", unknown, population mean (Field et al., 2012, p. 43ff). The standard CI has no direct
relationship to the parameter value nor is it based on its probability. It tells us about the probability of
data of "imaginary possibilities".

In contrast to that, the highest posterior density intervals (HPDI) cover the range of parameters (e.g.
regression weight) values that has a specified probability mass. If the parameter's probability to be
zero would be very high, the interval would cover mainly zero and very small values. In context of a
regression estimate in a linear model, this would indicate that changes in the predictor variable did not
result in changes in the outcome variable. This is not the same as the CI in the frequentist approach,
as uncertainty there is not related to parameter value probability, while the HPDIs are in fact true
probabilities of parameter values.

By reporting HPDIs, we describe the shape of the posterior probability distribution for our regression
parameters. The HPDIs are indicators about the range of parameter values compatible with the model,

the prior choice and the data (McElreath, 2020, p. 56 ff). They directly indicate the credibility of
regression coefficients and how consistent these values parameters are with the data, when applying
our design matrix and error structure (McElreath, 2020; Sorensen et al., 2016).

The reviewers further referred to the choice of priors and their influence on estimates. Inherently in
Bayesian analyses, HPDIs are influenced by the prior distribution, because the posterior results from
both, the prior and the likelihood, i.e., our data. Generally, priors encode states of information before
seeing the data and analyzing the shape of the posterior distribution is the best way to infer effects
based on observed data *and* prior knowledge.

Frequently many different priors produce the same inference and conventional Bayesian priors are
conservative relative to conventional non-Bayesian approaches. In our initial analysis we chose flat
uniform priors in the attempt to minimize the influence of the prior on the posterior distribution and
to let it be informed mainly by the observed data points following the approach in Sorensen et al.,
2016. This analysis resulted in the same effect pattern as an implementation of the equivalent models
in lme4 package in R, a widely used method to implement linear mixed effect models.

Recently, it has been advocated to use "weakly informative priors". The term "informative" is to be
understood in comparison to flat prior and actually renders the analysis more conservative as the
mode of the prior is centered on zero (Gabry et al., 2019). ([https://github.com/stan-
dev/stan/wiki/Prior-Choice-Recommendations](https://github.com/stan-dev/stan/wiki/Prior-Choice-Recommendations); Gelman et al., 2008).

As we consider the transition to Bayesian statistics very helpful in cognitive neuroscience, we wanted
to represent the current state-of-the-art and therefore refitted our models with weakly informative,
and therefore regularizing priors (intercept \sim normal(0,50); betas \sim cauchy (0,2.5) (Gelman et al.,
2008). We kept the uniform priors for the error terms, because a similar approach is outlined in
McElreath, 2020, p. 94-97 and we did not want to shrink the variance of the data. We report the new
estimates and HPDIs in the paper and conclusively for both samples (also see revised Supplemental
Table 2). Although the effects are slightly smaller as we applied a more conservative prior this did not
change the overall result pattern.

As a side note, we reran all analysis using the package lme4 in R (for results see Supplemental Table
4). With the use of additional packages (lmerTest, see below) it is possible to derive approximations
for p-values from those models. The conclusions from both our analyses converge and show that the
results with our new priors are more conservative than the results obtained with lme4. The model
equation included trial, session number, condition and intensity as fixed effects and the subject ID as
random intercept variable. Yet, we stick with our approach in the manuscript because it provides the
possibility to show the true probabilities of parameter values.

We also improved the visualization of the results. We changed the figures displaying rating data (Figure
1b,c and Figure S3a,b) and standard deviations (Figure 2a,b and Figure S5a,b) to avoid similarity in the
display of HPDI and/or parameter posteriors with frequentist confidence intervals. We decided to
display the entire posterior distribution of the estimated regression parameters as density curves for
the pairwise comparisons (SD model) or interactions (rating model). We highlight the position of the
zero on the scale, to show if probability curves of parameter value include zero. With the Bayesian
analysis approach, this distribution directly reflects the probability of the parameter value given our
data and the model.

Pp 4-5. It seems that statistics are reported on the behavioral-only sample, and then selectively for the
fMRI sample, without reporting full statistics for each sample in a way that clearly demarcates them.

We have included the parameter estimates for the interaction effects of pain ratings of both samples
in the main text of the manuscript. Additionally, we referenced the Supplemental Table 2 that contains
all model results for both samples regarding these ratings. Similarly, we provided the summary
statistics for the standard deviation analysis, including the mean and standard error for the conditions
and intensities. The most important regression parameters are reported in Figures 2 and S5, and we
have made sure to refer to them in the text.

Additional corrections: We corrected the reporting of the summary statistics of the standard deviations
of all conditions to standard errors not standard deviations of the standard deviations. Standard errors
were estimated with correction for within subject repeated measurements using Rmisc package that
employs the correction method from (Morey, 2008).

Finally (on the linear models), it's not clear what the random effects structure is for these models and
the degrees of freedom. Which effects were modeled as random? I don't believe random slopes for
experimental conditions are mentioned, but they would be appropriate here. It's difficult to interpret the
results without this information.

We chose linear mixed models as analysis tool, because these models allow to use all the data collected
in our study, without reducing it to summary values. In addition, these models allow to take the
repeated measures in the data into account and easily tackle a mix of continuous and categorical
predictor variables.

Concerning the random effect structure: We modelled the subject ID as a random effect, by including
random intercept for each subject, but did not introduce random slopes. We added intensity level and
condition as fixed effects, as well as trial and session numbers to control for habituation effects. We
rephrased the description in the Methods section and the main part of the paper, to clarify the
approach chosen and clearly mark all the fixed and random effects included in the model.

We did not include any random slopes, because their use in regard of all predictors of interest would
make the models very complex and we would risk non-convergence. Furthermore, we think that due
to our calibration procedure of intensities and targeting the individual VAS levels, slope differences
regarding the intensity levels between participants are accounted for.

The reviewers refer specifically to a random slope over experimental conditions, this would imply that
we would control for slope changes across intensities in the different conditions. But this is actually
our effect of interest, i.e. a steeper slope over intensities in condition C than in P. The slope differences
are included by modelling the pairwise interactions between the intensities and the dummy coded
conditions. Because an interaction is present (visible in the raw data) a main effect across conditions
would not be warranted and we do not separately report it.

As correctly stated by the reviewer, in general, the correct number of degrees on freedom in linear
mixed models is not known and can only be estimated (Kuznetsova et al., 2017). Packages like lme4 in
R therefore do not automatically report p-values, because it is unclear how the number of observations
should be determined. Packages like lmerTest apply an approximation, e.g. Satterthwaite, to derive
the degrees of freedom and *p*-values.

Given that we have adopted a full Bayesian approach, we do not have to assess t- or F-values resulting
from our models and do not have to compare them to a hypothetical distribution of values, with a
shape that is dependent on degrees of freedom.

**Analyses included and omitted**

I expected to see an analysis and report of the main effects of controllability compared with
unpredictable/uncontrollable and predictable compared with unpredictable/uncontrollable. It looks as
though there may be no effect of control overall, apart from providing an anchor for expectations. We
don't seem to be able to tell from this experiment whether control is analgesic, or assess the causal
effects of controllability on pain. This ambiguity is puzzling, as the brain contrasts analyze exactly these
kinds of main effects across all trials. So the behavioral data show bias towards predicted values, with
controllable and predictable conditions both offering strong value predictions, whereas the brain effects
show a different effect, the reduction (or increase) in pain-linked brain activity due to predictability and
controllability overall. It seems important to analyze both these overall (main) effects and interactions
(bias towards predicted levels) in both behavior and brain, and understand how they are linked.

**Behavior**

Concerning the behavioral data (ratings), we intentionally did not report main effects, because visual
inspection and later statistical model output indicated an interaction effect of intensity and condition.
If interactive patterns, that cross over one factor are present, it is difficult to interpret (or even invalid
to analyze) main effects, because this approach would make the data appear more consistent than
they are. Given these interactions we can conclude that the effect of control in our experiment is, in
fact, a stronger anchoring of expectations. We do not necessarily think that an interaction effect would
emerge in all types of studies investigating control effects on pain, particularly if only one intensity
level is examined. However, the increase in expectation precision could be relevant for all types of
control, with respect to the feature of the stimulus over which control is exerted. For example, when
the onset is controllable, its start can become more predictable; when its intensity is controllable, its
intensity can become more predictable.

Nevertheless, we show the averaged condition ratings for each sample (see Figure E). These plots
should be interpreted with caution due to the interaction discussed above. The average means of all
conditions are very similar; however, there is a small main effect among the conditions, although this
effect is inconsistent across samples (the controllable condition is rated higher in sample 1 (behavioral)
than the predictable condition, but lower in sample 2 (fMRI)). The unpredictable condition resulted in
slightly higher ratings in both samples. We believe that this effect arises from the fact that the
differences between the conditions at low intensity levels were larger than at high intensity levels.

**Figure E** | Raw ratings and distribution of ratings with indication of median (bar in box), and mean (point and
numerical value). Values are averaged across three intensity levels. Because intensity level and condition interact
this plot needs to be interpreted with caution.

**Brain**

To provide the equivalent analysis for fMRI data that we report for behavior, we tested for interactions
between intensity levels and condition in brain activity during pain. To implement this analysis, we set
up a GLM with separate pain onset regressors for each condition and added the (z-scored) intensity

levels to each pain onset regressor as a parametric modulator. We then tested all pairwise interactions
between the conditions.

At the whole brain FWE corrected significance threshold (at $p < .05$), no voxel was significant for any of
the interaction contrasts. Nevertheless, visual inspection of the interaction effects at an uncorrected
threshold of $p < .001$ revealed multiple activation clusters. In particular, we found that the activity in a
cluster located in the rostral ACC (interaction between controllable and unpredictable), showed the
same pattern as the pain ratings, i.e. activity in this ACC cluster positively scales with intensity in the
controllable condition and negatively with intensity in the unpredictable condition (see Figure Fb).
Although this is clearly a post-hoc analysis, we find this interesting, because the rostral ACC is a region
known to encode expectation-sensation integration of pain. To put this in perspective (again post-hoc),
this activation would be significant using an ACC mask and small-volume correction (p (small volume
corrected) = .003 in voxel = [-7.5, 37.5, -1.5]; $T = 4.96$). However, we would like to stress that this is a
post-hoc small volume correction and only a trend at the whole brain FWE corrected level. We suggest
that this could be interesting and might inform future studies on the topic. Therefore, we decided to
mention this result, including a figure (see below) in the Supplemental Information (but not in the
discussion) in the revised manuscript. However, given the post-hoc nature of this analysis, we could
understand that the reviewers suggest omitting this result.

Conclusion

We showed that if averaged across intensities, the condition differences in ratings become very small,
as this analysis obscures the underlying interaction. Regarding the analysis of condition main effects in
brain activity, we think that our original analysis approach is still valid, because these condition
dependent brain activity changes might in fact be related to different levels of expectation precision
rather than the pain percept that is reflected in the ratings. The main effects in fMRI are therefore
closer to the analysis of standard deviation differences between the conditions than to the analysis of
mean ratings.

**Figure F** [(a) Thresholded statistical maps for the interaction effect between controllable and unpredictable
 condition, masked with an ACC-mask, which was also used for (post-hoc) small volume correction. Green areas
 indicate regions significant at uncorrected $p < .001$. Pink region show the cluster that was significant with small
 volume correction. (b) Parameter estimates of the significant interaction voxel, visualized with a 9-regressors
 model separately showing activity in the voxel for the three different task conditions at low (L), medium (M) and
 high (H) stimulus intensity. While there is a positive scaling with intensity for controllable and predictable
 condition, activity decreases with intensity in the unpredictable condition.

The main behavioral finding is that the standard deviations of ratings were lower in the
controllable condition than in the predictable condition in the fMRI sample. The authors
attribute this to an effect of control, as the stimulus intensities are nominally the same. But
inferential statistics are not provided.

We ran linear mixed models on the standard deviations of ratings and compared the differences across
conditions and intensities using a similar approach as that for the raw rating data. A description of the
model and results can be found in the Methods section, as well as in Figures 2 and S5. Supplemental
Table 3 in the supplementary material summarize the regression estimates and HPDI for the analysis
of standard deviations of both study samples

The analyses assume the mean shift or precision change is consistent (the same mechanism) in
both tasks. What if one task has mean shift mechanism and the other has precision modulation?
Also, what if both mechanisms are involved? Is it possible to expand the model comparisons to
test these possibilities?

Since we can only measure parameter changes relative to another condition, it is important to note
that changes in mean and precision are relative to the specific conditions being compared. For the
differences between the *unpredictable* condition and the two other conditions, our expectation-
integration model indeed assumes a mean-shift, because participants' best guess about the
painfulness is the expected value, i.e. the average of all possible intensities for pain, while the intensity
level is certain in the other two conditions. We believe it is unlikely that the pairwise differences
between the controllable and unpredictable conditions, or between the predictable and unpredictable
conditions, can be solely attributed to mean shifts or changes in precision. This is because the
imprecise information characteristic of the unpredictable condition suggests that there would be a
different expected mean *and* higher variance.

Regarding the differences between the controllable and predictable condition, we aimed to determine
which mechanism better explains the observed data patterns. Our findings indicate that changes in
precision provide a better explanation of the data than assuming different means, because our
modelling analysis showed that the differences in precision were more pronounced than any shifts in
mean, accounting for the rating differences.

While our analysis does not rule out the possibility that both mechanisms may contribute to the
differences observed between conditions, we have shown that the effect of precision differences is
more substantial.

The analyses assume that shifts in mean and variance (precision) are independent, and that a
mean shift will entail no change in the variance of its associated distribution. However, this is
a strong assumption that may not be warranted. The means and variances of measured
phenomena are often linked.

If so, the differences between the controllable and predictable conditions could be due to
differences in mean levels for controllable and uncontrollable rather than qualitative differences
in precision.

The standard deviation, which is used to estimate precision, varies with the mean and cannot
be assumed to be constant across the range of the rating scale.

Can you provide evidence that the mean and precision are independent here, and potentially
explore models in which they are not strongly decoupled?

The reviewers suggest that variations in the mean could lead to changes in standard deviations, as
these two parameters may be related and the standard deviation might not remain constant across
the rating scale. In response to this concern, we have conducted analyses to examine the relationship
between mean ratings and standard deviations across conditions and the rating scale in our data.

The standard deviation was smallest in ratings in the controllable condition, followed by predictable
and unpredictable condition in sample 2, and smaller in the controllable and predictable condition
than in the unpredictable condition in sample 1, across all intensity levels (see Figure G). Assuming a
linear relationship of standard deviation and mean rating this pattern would be incongruent with the
interaction effect, which is present in the mean rating data.

**Figure G** | Average standard deviations grouped for condition and intensity levels for both samples. Condition
differences are present at all intensity levels. sample 1 = behavioral sample; sample 2 = fMRI sample.

However, as inferred from the analysis of standard deviations across different intensity levels (see
Figures 2 and S5), the within-subject standard deviations exhibit a quadratic relationship with the
mean ratings of the participants. While the relationship between the mean and standard deviation is
positive for the low and medium intensity levels, it becomes negative for the high intensity level, with
a general offset between the conditions (see also Figure H).

Taking the interaction effect of mean ratings across intensity levels and conditions into account and if
looking at the data "model-free" without any mechanistic explanation, we could state that there is
indeed a coupling between both parameters, because lower mean ratings for low intensity stimuli in
the controllable condition along with a lower standard deviation and higher mean in the controllable
condition for the high intensity level along with a lower standard deviation, are in line with the
observed inverted U-shape pattern (although this could not explain the results at the medium intensity
level).

As a matter of fact, we do not expect the mean and the precision of the ratings to be completely
independent. Our model predicts that different precisions of expectation result in a difference in the
posterior mean (of course also depending on the prior mean).

Because the rating mean (i.e. posterior mean) is defined as the integration of the *precision-weighted*
expectation and the *precision-weighted* sensory input and the rating precision is the summed precision
of expectation and sensory input, a change in posterior mean can be accompanied by changes in the
precision. A different mean between the conditions on the other hand, at least in our theoretical
approach cannot explain differences in posterior precision.

**Figure H |** Within subject standard deviations plotted against mean ratings in all three conditions and intensity
levels with fitted lines (both samples). Standard deviations scale positively with rating mean at low and medium
stimulus activity and negatively with rating mean at high intensity level. sample 1 = behavioral sample; sample 2
= fMRI sample.

What features of the data are driving the differences in model fit between the mean shift and
 precision shift models? It is not clear how different the predictions of the two models are in this
 case. This is interesting as the results appear to conflict with the type of effect found in similar
 recent work by Strube et al. from the same lab. It is also important to understand because of the
 approximations used here (both use of Gaussian distributions and approximate matching of the
 stimulus intensity and sequence of intensities) and the potential for shifts in mean ratings to be
 linked to shifts in the variance of ratings in ways the models do not account for.

The feature that drives the difference in model fit, are the differences in standard deviations between
 the conditions, especially because they are more substantial than the differences in means.

Model comparison indicated that the model that expected only the mean values to differ between the
 conditions was less likely to have produced the data pattern than the model that allowed different
 standard deviations of ratings. While the predictions for the mean values of ratings are not very
 different between the two models, the different predictions for the standard deviations were the
 driving factor behind the better model fit of the precision change model (see Figure I, comparing mean-
 shift model, precision change model and the rating data).

We think there are multiple points that are relevant in comparison to the work by Strube et al, that
 found an impact of agency resulting in a unidirectional prior mean shift. These are discussed in detail
 below (see Theoretical implications).

 **Figure I** | Predicted and empirical ratings and the corresponding SD for mean-shift and precision-change model
 and ratings for all three conditions.

**Theoretical implications**

The findings are cast in terms of validating Bayesian theories of pain, but the authors may want to word
their intro and discussion carefully, because it is not clear that the Bayesian model makes strong
predictions here that are not made by other theories, and thus the paper doesn't really "prove" that the
Bayesian model is the "right" one. Existing theories in psychology suggest that expectations influence
perception, causing biases in perception towards predicted values (or other sources of information),
which produces the kinds of interaction effects found here.

Even simply adding more noise in judgments in the unpredictable condition (and in the predictable
relative to the controllable condition) would produce the same qualitative pattern of interactions.

The paper also doesn't attempt to compare Bayesian with non-Bayesian models. The Bayesian
framework is very useful and sensible, but I would feel more comfortable if these results were not
presented as a strong validation of Bayesian theories.

As suggested by the reviewers, we applied the Bayesian framework because it is "useful". The main
focus of the paper was not to prove the Bayesian model right, but to explore different mechanism that
led to a change of pain ratings if the stimulus intensity was self-chosen. The Bayesian approach was
considered useful, because it has been successfully applied to other studies (Geuter et al., 2017; Grahl
et al., 2018; Strube et al., 2023) and similar mechanisms have been suggested in many other sensory
domains (Petzschner et al., 2015).

For application of alternative *non*-Bayesian models, the question would be what aspect of the Bayesian
model should be compared to a non-Bayesian one and what kinds of non-Bayesian models should be
tested. We call the applied models *Bayesian*, because they formalize the pain percept as *precision-*
*weighted* integration of a hypothesized distribution of *expectation* and the applied sensory input; both
pivotal features of Bayesian models. We think that the expectation plays a crucial role, and by including
and comparing our expectation-integration models with a null model that did not include differences
in expectation, we think that we indeed performed a comparison to a non-Bayesian model in that
regard. The other Bayesian aspect of our models, the precision-weighting, is explicitly formalized in
the unpredictable condition and applied to the interpretation of the data and results of all conditions.
We think that the differences in the standard deviation of ratings support this explanation.

This paper seems to be inconsistent at a theoretical level with Strube and Büchel's work from the same
lab. There, they test an overlapping construct, agency, and conclude that it results from a shift in priors
(lower pain expectations). They test a likelihood precision model, positing that even though the stimulus
intensities are the same for agency and no-agency conditions, the likelihood precision may differ.

Here, however, it is claimed that the likelihoods must in principle be the same across control and no-
control conditions because the stimulus intensities are the same. This seems like a strong and
unwarranted assumption given the prior work.

We do not think that is necessarily an unwarranted assumption given the prior work. The likelihood in
the many Bayesian models of (pain) perception is often referred to as "somatosensation" (Strube et
al., 2023), "incoming nociceptive information" (Grahl et al., 2018), "incoming sensory data" (Büchel et
al., 2014) or "noisy sensory input" (Petzschner et al., 2015). It is thus not unwarranted to assume that
the likelihood is the same across conditions, given that the same temperatures were applied and that
our model takes habituation/sensitization effects into account. Please also note that the protected
exceedance probability of the likelihood precision model in the study by Strube was extremely low
(<1% in both studies).

In general, it is a theoretical question at what level of sensory processing the likelihood should be
inferred in the context of pain and this might also depend on the experimental design. Because we
chose to individually calibrate the temperature and always applied the same intensities in all
conditions, we consider it valid to assume that apart from habituation effects "the incoming sensory
data" was kept constant throughout the conditions. Depending on one's definition of "likelihood" it is
in theory conceivable that the likelihood precision can change across conditions, but with our
experimental setup we consider it more likely that these changes affect the expectation component
of the model and not the sensory part. This is not in conflict with other studies that assume differential
influence of their conditions on lower order sensory level, like in application of the active inference
model in (Strube et al., 2023) as outlined further in the discussion below.

It seems possible that the experiments do not have the necessary conditions to disentangle the various
possibilities.

With the data at hand, a full Bayesian model, where likelihood and prior parameters (mean and
precision) are free to vary without any constraints, would not be identifiable because the parameters
depend on each other. We still wanted to provide more insights into the results and approximated the
precision of the prior by the variance of the samples in the posterior and implemented the alpha
parameter as a proxy for the variance ratio. We do not think that this is problematic, as models "*are*
*always wrong but sometimes useful*" (George Box) and information reduction is important when trying
to test parts of a hypothesis to increase understanding.

After discovering the change in rating consistency (standard deviations) between the conditions and
the interaction effect of the means, we considered it reasonable to assess if changes in expectation
precision could help us with the interpretation of the rating differences between the controllable and
predictable conditions. We focused on the prior precision, because we saw a similar effect pattern
when comparing the predictable vs. controllable and the predictable vs. unpredictable condition
(where the expectation was inherently less precise). This convinced us that changes in expectation
precision would be a likely candidate to explain also differences between controllable and predictable
condition and provided us with interesting results even though not all possible variations of model
component changes could be tested.

For example, a higher prior precision with control could in fact be a lower posterior precision, and this
may explain the same effects. If such ambiguities are present, it would be helpful to explain them
transparently.

1) *We guess that the reviewers wanted to refer to "lower likelihood precision" and not "lower posterior*
*precision". We see the smaller standard deviation in ratings in the controllable condition than in the*
*two other conditions as an indirect indicator for a higher posterior precision and have interpreted this*
*finding in light of increased prior precision, or, subjective expectation acuity. However, the reviewer is*
*right that mathematically a difference in posterior precision between conditions could, in theory,*
*either stem from differences in prior precision or likelihood precision or a combination of both.*

Given our raw data, a model that implemented different likelihood precisions, while keeping the
likelihood mean and the prior distribution constant across conditions, would result in a higher
likelihood precision parameter in controllable condition, if the corresponding shared prior distribution
would be centered at the likelihood mean or higher for low stimuli and at the likelihood mean or lower
at high intensities in order to explain the interaction. It follows that sensory acuity must be *higher* in
the controllable condition, and lower in the predictable condition and that the expectation precision
is equal in both. However, the notion that people have higher sensory acuity, when they are in control
is the opposite of what *active inference theory*, tested in Strube et al. 2023 would suggest. To further
illustrate this point we have performed a simulation of this outcome. As can be seen in the figure
below, this would yield the opposite result pattern as compared to our observed pain ratings.

**Figure J |** Simulated ratings (points) and averages (bars) for controllable (dark green) and predictable (light green)
conditions with constant priors and a lower likelihood precision in the controllable condition (what would be
suggested by active inference theory). This would yield the opposite result pattern as compared to our pain rating
data.

We think that changes of sensory acuity are even less probable in our setup than in the one applied in
Strube et al., because in the previous study, participants received immediate sensory feedback about

the self-administered or externally induced pain reduction, whereas in our task there was a longer
temporal delay between the decision on intensity and the actual application of the stimulus. Agency is
often discussed in the framework of sensorimotor feedback and voluntary movements and here
testing for differences in sensory acuity is warranted, while with larger temporal gap and a more
cognitive type of subjective control, this becomes less relevant. We would not assume that the decision
on a temperature that a device will apply with a long delay after the decision, as in our design, would
lead to sensory acuity reduction. We think that this influenced rather the shape of expectations than
sensation.

2) As pointed out by the reviewers, Strube et al. 2023 report that agency led to different pain
perception due to a shift in *prior means* instead of a change in likelihood precision, while our modelling
results suggest a change in *prior precisions*. We think that there are multiple explanations for the
differences in modelling results.

In Strube, 2023 pain was reduced from a high pain level (VAS 70) to a relatively low intensity (VAS 30)
after participants got conditioned to expect either higher or lower treatment efficacy. In one condition,
participants were led to believe that they initiated the (sham) treatment, while in the other condition,
the experimenter started the treatment. The self-initiated treatment led to higher pain reduction than
the externally initiated treatment for both expectation condition, indicating a general downwards shift
of expected pain (prior mean).

A unidirectional mean shift for self-chosen stimuli is not evident from our data pattern. Only an
inconsistent change of prior mean by control (i.e. to be higher for high pain and lower for low pain)
could explain the result pattern. This interaction effect can be parsimoniously explained by a change
in expectation precision, but not a shift of the prior means, and this was backed up by our model
comparison. Additionally, it would be difficult to theoretically motivate why a prior mean shift would
have different directions for low and high pain than to (parsimoniously) assume that control increases
subjective expectation precision.

Even though results of both studies rely on expectation and sensation integration, we find that they
address different aspects of pain modulation by agency and control. In case of Strube 2023, all
conditions entailed a pain reduction compared to a peak at the beginning of the trial, while in our case,
intensity was never reduced but increased. The cover story of the Strube study is a treatment context,
while our study addressed perceptual differences regarding predicted, unpredicted and self-chosen
pain intensities. It is plausible that people do not shift their expectation if they only have control over
different intensities, which are all painful; but do so if they are being able to exert control over
something positive, like treatment onset, and that controlling something positive might reduce the
expected painfulness in general.

Finally, we think that these two studies are an excellent example for different ways of thinking about
effects of cognitive factors on pain and pain relief. Together they illustrate a broader picture of control
effects on pain and that results might differ with regard to operationalization of control, direction of
self-controlled pain modulation (control over pain offset, or reduction like in Strube; control over pain
intensity and increase like in this paper), changes in temporal aspects of stimulation (immediate vs.
delayed) or experimental framings (no placebo instruction or deception vs. placebo conditioning,
treatment context).

The alpha parameter seems to be a non-Bayesian replacement for the precision of the stimulus, and it is
unusual in that it seems to be outside the Bayesian framework the paper purports to test. Why not actually
parameterize the precision of the stimulus (or relative precision of prior and stimulus)?

As mentioned above, a full Bayesian model, where likelihood and prior parameters (mean and
precision) are free to vary without any constraints, would not be identifiable in our case, because the
parameters are not fully independent. We tested the model with random parameters for prior and
stimulus precision, but as we do not have data regarding the expectation distribution of participants
these parameters were not identifiable, because, as a ratio (prior variance/likelihood variance + prior
variance; see BDA3, page 50 (Gelman, 2014)) either the one or the other could vary.

Therefore, and as usual in computational modelling, we decided to express this ratio as one parameter
that parametrizes the weight given to the likelihood mean (calibrated value). Consequently, the alpha
parameter implements the relative precision of prior and stimulus in the unpredictable condition. For
the other conditions, we sample directly from the prior, based on our assumption of constant
likelihood. We tested models that implemented separate alpha parameters for all conditions, but
parameters of this model were not recoverable given the data at hand, as we discussed in the Methods
section of the manuscript.

**Other clarifications**

The authors suggest that control increases outcome certainty, more than mere predictability. The
distinction between certainty and predictability, while mentioned, could benefit from additional
clarification for non-specialist readers. Further, in line with the first point that was raised, I am not
convinced that controllability increases outcome certainty in general – though it does in this paradigm -
- especially if the outcomes are probabilistic and influenced by multiple factors. It may be helpful to
discuss that controllability may be differentially associated with predictability across different contexts.

This might be a misunderstanding. It is important to note that only the features of the stimulus that
are genuinely under control become more predictable. For instance, if the offset of a stimulus can be
controlled, this does not necessarily lead to increased certainty about its intensity. Conversely, if the
intensity is under control, one cannot assume that there is heightened certainty regarding the duration
of the stimulus.

We devoted several sentences in the discussion to this distinction and emphasized the broader nature
of certainty, as well as the potentially different effects that can arise in various contexts of
controllability and predictability. We also related our findings to effects of (motor) agency previously
mentioned, particularly in reference to the results from the study by Strube et al.

The behavioral findings on predictability are described as though predictability causes a complex effect
of decreases in low-intensity and increases in high-intensity stimuli (see Results and Discussion). While
technically an interaction effect, the language obscures the likely story by casting predictability as a
causal construct. A more parsimonious explanation is that ratings are biased towards their predicted
values, so if a person expects a low-intensity stimulus with high confidence (high precision), they report
lower intensity, and if a person expects a higher-intensity stimulus, they report higher intensity. I think
the authors likely agree with this explanation, as it fits the Bayesian framework, but the writing could
tell this story more clearly.

We would like to thank the reviewer for their feedback that nicely captures what we intended to
communicate. We have now refined the phrasing throughout the Results and Discussion so that it
aligns with our intended message.

**Other points**

PAG and aversive prediction errors: Is it unpredictability and/or lack of control that increases
PAG activity, or is it worse-than-expected stimuli (aversive prediction errors) specifically?

With our current setup, we are unable to determine the participants' expectations and the precise
prediction errors at the trial level, as we can only obtain average readouts across trials. Consequently,
we cannot specifically identify stimuli that are perceived as particularly worse than expected without
making additional assumptions.

In Fig 5 C/D it does not look like any regions survived multiple comparisons, as there are no
areas in black outlines as in 5A/B?

There are regions that survived correction for multiple comparisons, but the linewidth used before
rendered the area to look like a point. We adjusted linewidth in the Figure 5c,d and show a zoomed in
display to make the voxels that survives correction for multiple comparisons more visible.

Please describe the anchors used for VAS ratings (e.g., of 0 and 100), and summarize with
statistics the stimulus intensity for threshold and tolerance, and how these map onto VAS
ratings.

The only anchors for the VAS scale were 0 and 100, corresponding to the label of "minimal pain" and
"unbearable pain" respectively. Participants were instructed, that unbearable pain was to be
understood in context of their participation in the experiment. Before the calibration procedure,
participants were familiarized with the scale and were verbally instructed about what the different
segments of the scale were related to. Numerical values were never shown or mentioned to
participants.

We refer the reviewers to the tables below for the stimulus intensities in °C of predicted threshold
(VAS 0) and tolerance (VAS 100) of the participants and the intensities that were actually applied in
the experiment (VAS 30, 50 and 70) for both samples respectively. Distribution of data is shown in the
corresponding boxplots for each VAS level. The we added the VAS 0 and VAS 100 intensity levels also
to the Supplemental Table 6.

Stimulus intensities (behavioral sample)

	VAS 0	VAS 30	VAS 50	VAS 70	VAS 100
Temp in °C M (SD)	40.93 (4.25)	43.56 (2.63)	45.31 (1.66)	47.07 (1.14)	49.76 (2.05)

Stimulus intensities (fMRI sample)

	VAS 0	VAS 30	VAS 50	VAS 70	VAS 100
Temp in °C M (SD)	42.09 (2.86)	44.31 (2.06)	45.79 (1.67)	47.2 (1.39)	49.5 (1.81)

Figure K | Distributions of calibrated stimulus intensities of VAS 0 (corresponding to pain threshold), VAS 100 (corresponding to pain tolerance) and of the VAS targets applied throughout the tasks (VAS 30, VAS 50, VAS 70).

**References**

- Barron, H. C., Garvert, M. M., & Behrens, T. E. J. (2016). Repetition suppression: A means to index
neural representations using BOLD? *Philosophical Transactions of the Royal Society B:*
*Biological Sciences*, 371(1705), 20150355. <https://doi.org/10.1098/rstb.2015.0355>
- Boly, M., Baetens, E., Schnakers, C., Degueldre, C., Moonen, G., Luxen, A., Phillips, C., Peigneux, P.,
Maquet, P., & Laureys, S. (2007). Baseline brain activity fluctuations predict somatosensory
perception in humans. *Proceedings of the National Academy of Sciences*, 104(29), 12187–
12192. <https://doi.org/10.1073/pnas.0611404104>
- Büchel, C., Geuter, S., Sprenger, C., & Eippert, F. (2014). Placebo Analgesia: A Predictive Coding
Perspective. *Neuron*, 81(6), 1223–1239. <https://doi.org/10.1016/j.neuron.2014.02.042>
- Field, A., Miles, J., & Field, Z. (2012). *Discovering statistics using R*. Sage.
- Fritsche, M., Lawrence, S. J. D., & De Lange, F. P. (2020). Temporal tuning of repetition suppression
across the visual cortex. *Journal of Neurophysiology*, 123(1), 224–233.
<https://doi.org/10.1152/jn.00582.2019>
- Gabry, J., Simpson, D., Vehtari, A., Betancourt, M., & Gelman, A. (2019). Visualization in Bayesian
Workflow. *Journal of the Royal Statistical Society Series A: Statistics in Society*, 182(2), 389–
402. <https://doi.org/10.1111/rssa.12378>
- Gelman, A. (2014). *Bayesian data analysis* (Third edition). CRC Press.
- Gelman, A., Jakulin, A., Pittau, M. G., & Su, Y.-S. (2008). A weakly informative default prior
distribution for logistic and other regression models. *The Annals of Applied Statistics*, 2(4).
<https://doi.org/10.1214/08-AOAS191>
- Geuter, S., Boll, S., Eippert, F., & Büchel, C. (2017). Functional dissociation of stimulus intensity
encoding and predictive coding of pain in the insula. *eLife*, 6, e24770.
<https://doi.org/10.7554/eLife.24770>
- Grahl, A., Onat, S., & Büchel, C. (2018). The periaqueductal gray and Bayesian integration in placebo
analgesia. *eLife*, 7, e32930. <https://doi.org/10.7554/eLife.32930>

Huang, Y., & Rao, R. P. N. (2011). Predictive coding. *WIREs Cognitive Science*, 2(5), 580–593.
<https://doi.org/10.1002/wcs.142>

Klein-Flügge, M. C., Barron, H. C., Brodersen, K. H., Dolan, R. J., & Behrens, T. E. J. (2013). Segregated
Encoding of Reward–Identity and Stimulus–Reward Associations in Human Orbitofrontal
Cortex. *The Journal of Neuroscience*, 33(7), 3202–3211.
<https://doi.org/10.1523/JNEUROSCI.2532-12.2013>

Kuznetsova, A., Brockhoff, P. B., & Christensen, R. H. B. (2017). **lmerTest** Package: Tests in Linear
Mixed Effects Models. *Journal of Statistical Software*, 82(13).
<https://doi.org/10.18637/jss.v082.i13>

McElreath, R. (2020). *Statistical rethinking: A Bayesian course with examples in R and Stan* (2nd ed.).
Taylor and Francis, CRC Press.

Morey, R. D. (2008). Confidence Intervals from Normalized Data: A correction to Cousineau (2005).
*Tutorials in Quantitative Methods for Psychology*, 4(2), 61–64.
<https://doi.org/10.20982/tqmp.04.2.p061>

Petzschner, F. H., Glasauer, S., & Stephan, K. E. (2015). A Bayesian perspective on magnitude
estimation. *Trends in Cognitive Sciences*, 19(5), 285–293.
<https://doi.org/10.1016/j.tics.2015.03.002>

Segaert, K., Weber, K., De Lange, F. P., Petersson, K. M., & Hagoort, P. (2013). The suppression of
repetition enhancement: A review of fMRI studies. *Neuropsychologia*, 51(1), 59–66.
<https://doi.org/10.1016/j.neuropsychologia.2012.11.006>

Sorensen, T., Hohenstein, S., & Vasishth, S. (2016). Bayesian linear mixed models using Stan: A
tutorial for psychologists, linguists, and cognitive scientists. *The Quantitative Methods for*
*Psychology*, 12(3), 175–200. <https://doi.org/10.20982/tqmp.12.3.p175>

Story, G. W., Vlaev, I., Dayan, P., Seymour, B., Darzi, A., & Dolan, R. J. (2015). Anticipation and Choice
Heuristics in the Dynamic Consumption of Pain Relief. *PLOS Computational Biology*, 11(3),
e1004030. <https://doi.org/10.1371/journal.pcbi.1004030>

Story, G. W., Vlaev, I., Seymour, B., Winston, J. S., Darzi, A., & Dolan, R. J. (2013). Dread and the
Disvalue of Future Pain. *PLoS Computational Biology*, 9(11), e1003335.
<https://doi.org/10.1371/journal.pcbi.1003335>

Strube, A., Horing, B., Rose, M., & Büchel, C. (2023). Agency affects pain inference through prior shift
as opposed to likelihood precision modulation in a Bayesian pain model. *Neuron*,
S0896627323000028. <https://doi.org/10.1016/j.neuron.2023.01.002>

REVIEWERS' COMMENTS

Reviewer #1 (Remarks to the Author):

The authors have done an outstanding job in addressing the reviewers' comments. They not only provided additional information but also conducted further analyses and included new figures, all of which are highly convincing.

However, a more fundamental scientific question remains in the background—though it does not detract from my overall positive assessment of the manuscript. This pertains to the added value of the behavioral versus the imaging results. While it is reassuring that the fMRI findings align with the hypothesis (which is also supported by the behavioral data) that control effects are associated with increased expectation precision, the authors' response—"We think that the imaging data is crucial to validate the model by bringing forward the neurophysiological processes possibly underlying the proposed mechanisms"—does not entirely resolve my concerns. Specifically, had the imaging data not fully corroborated the behavioral results, how problematic would this discrepancy have been?

This issue brings to mind Gregory Miller's (2010) seminal paper, which eloquently discusses the challenge of defining what constitutes an adequate explanation in the relationship between psychological and biological phenomena. The paper cautions against assuming direct causal links between neural and psychological events, as they operate at different levels of analysis. If they agree, the authors may wish to add a sentence or two in the discussion section of the manuscript.

Answer:

We thank the reviewer for their overall positive assessment of our article and for the time and effort they invested in the review.

Additionally, we would like to thank the reviewer for the comment touching on an important and intrinsic problem in human neuroscientific research. In response, we added the following sentence in the discussion:

"The fMRI results fit the mechanism that was also put forward by our modeling analysis of behavior. Nevertheless, the exact mapping between changes in BOLD signal and psychological effects is often not known and it is therefore difficult to establish a direct causal relationship between both levels of analysis."

Reviewer #1 (Remarks on code availability):

N.A.

Reviewer #4 (Remarks to the Author):

I am reviewing the revised manuscript on behalf of Reviewer #2, who provided a comprehensive review of the original manuscript but is currently unavailable for further comments. I concur with Reviewer #2 regarding the value of the study, particularly the observation that "the study makes an interesting and novel claim" and that comparing controllable versus uncontrollable conditions while matching on predictability is "an innovative" approach. The topic is indeed "a perennially important one," as noted.

Reviewer #2's comments were thorough, and some of them touched on fundamental issues. However, I find that the authors have responded to all points both elegantly and comprehensively. They conducted multiple additional analyses, which have been included in the supplementary information. In my view, the reviewer's feedback and the authors' revisions have substantially improved the manuscript. I therefore recommend that the paper be accepted in its current form.

Answer:

We thank the reviewer for their overall positive assessment of our article and for the time and effort they invested in reviewing the revisions made in response to the comments from the previous review team. We sincerely appreciate your time and effort!